# Atmospheric CO2, CH4, and CO with CRDS technique at the Izaña Global GAW station: instrumental tests, developments and first measurement results

Angel J. Gomez-Pelaez[1,a], Ramon Ramos[1], Emilio Cuevas[1], Vanessa Gomez-Trueba[1,2], Enrique Reyes[1]

[1]Izaña Atmospheric Research Centre, Meteorological State Agency of Spain (AEMET), Tenerife, Spain.
[2]Air Liquide España, Delegación Canarias, Tenerife, Spain.
[a]now at: Meteorological State Agency of Spain (AEMET), Delegation in Asturias, Oviedo, Spain.

*Correspondence to*: Angel J. Gomez-Pelaez (agomezp@aemet.es)

**Abstract.** At the end of 2015, a CO2/CH4/CO Cavity Ring-Down Spectrometer (CRDS) was installed at the Izaña Global Atmosphere Watch (GAW) station (Tenerife, Spain) to improve the Izaña Greenhouse gases GAW measurement programme, and to guarantee the renewal of the instrumentation and the long-term maintenance of this programme. We present the results of the CRDS acceptance tests, the raw data processing scheme applied, and the response functions used. Also, the calibration results, the implemented water vapour correction, the target gas injection statistics, the ambient measurements performed from December 2015 to July 2017, and their comparison with other continuous in situ measurements are described. The agreement with other in situ continuous measurements is good most of the time for CO2 and CH4, but for CO it is just outside the GAW 2-ppb objective. It seems the disagreement is not produced by significant drifts in the CRDS CO World-Meteorological-Organization (WMO) tertiary standards. The more relevant contributions of the present article are: 1) determination of linear relationships between flow rate, CRDS inlet pressure and CRDS outlet valve aperture; 2) determination of a slight CO2 correction that takes into account changes in the inlet pressure/flow rate (as well as its stability over the years), and attributing it to the existence of a small spatial inhomogeneity in the pressure field inside the CRDS cavity due to the gas dynamics; 3) drift rate determination for the pressure and temperature sensors located inside the CRDS cavity from the CO2 and CH4 response function drift trends; 4) the determination of the H2O correction for CO has been performed using raw spectral peak data instead of the raw CO provided by the CRDS and using a running mean to smooth random noise in a long water-droplet test (12 hours) before performing the least square fit; and 5) the existence of a small H2O dependence in the CRDS-flow and of a small spatial inhomogeneity in the temperature field inside the CRDS cavity are pointed out and their origin discussed.

## 1 Introduction

A CO2/CH4/CO Cavity Ring-Down Spectrometer (CRDS) was installed at the Izaña Global Atmosphere Watch (GAW) station (Tenerife, Spain) at the end of 2015 in order to improve the Izaña Greenhouse-gas (GHG) GAW measurement

programme, and guarantee the long-term maintenance of this programme. The incorporation of the CRDS technique for the measurement of atmospheric CO2, CH4, and CO mole fractions was a recommendation of the World Meteorological Organization (WMO) World Calibration Centre (WCC) EMPA after its audit carried out at the Izaña Observatory (IZO) in September 2013 (EMPA, 2013).

The WMO GAW programme requires high precision and accuracy in atmospheric GHG measurements, which are more stringent for trace gases with a longer lifetime in the atmosphere. The reason is that atmospheric GHG spatial gradients contain useful information about the spatial distribution of the surface sources and sinks of these trace gases (Chevalier et al., 2010), but these gradients decrease in absolute value as the trace gas lifetime increases (Patra et al., 2014). The GAW required compatibility between laboratories is 0.1 ppm (parts per million in dry mole fraction; i.e., micromoles per mole of

dry air) for carbon dioxide in the Northern Hemisphere and 0.05 ppm in the Southern Hemisphere; for methane, 2 ppb (parts per billion in dry mole fraction; i.e., nanomoles per mole of dry air); and for carbon monoxide, 2 ppb (WMO, 2015).

The CRDS technique (Crosson, 2008) has improved considerably the stability and precision in the raw measurements compared to those of older techniques (e.g., Non Dispersive InfraRed analysers -NDIRs-, Gas Chromatography -GC- with Flame Ionization Detector -FID-, and GC with Reduction Gas Detector -RGD-), therefore, the required frequency of use of

calibrating/reference gases to achieve the GAW Data Quality Objectives (DQOs) is much lower. Additionally, this spectrometric technique does not require chromatographic gases (e.g., carrier gas, make up gas, and FID gases for maintaining the flame), which are expensive, and require great logistics efforts at remote stations. Zellweger et al. (2016) found out, when evaluating the results of scientific audit performed at GAW stations, that the results using newer spectroscopic techniques (CRDS and Off-Axis Integrated Cavity Output Spectrometry -OAICOS-), in general, are better

than those obtained with older techniques.

The Izaña Observatory is a Global GAW station located at 2373 metres a.s.l. on Tenerife (Canary Islands, Spain), usually above a strong subtropical temperature inversion. Since it is located at the summit of a mountain, during night-time there are North-Atlantic-free-troposphere background conditions, whereas during daytime there is a slight perturbation of these conditions by the arrival of an upslope thermal wind close to the terrain surface (e.g., Cuevas et al., 2013; Gomez-Pelaez et

al., 2013; Rodríguez et al., 2009). Detailed information about the numerous measurement programmes in operation at IZO are provided by Cuevas et al. (2015).

This paper presents the implementation of a CRDS G2401 at IZO and the development of the raw data processing scheme. The more relevant contributions of the present article are: 1) determining linear relationships between flow rate, CRDS inlet pressure and CRDS outlet valve aperture; 2) determining a slight CO2 correction that takes into account changes in the inlet

pressure/flow rate (as well as its stability over the years), and attributing it to the existence of a small spatial inhomogeneity in the pressure field inside the CRDS cavity due to the gas dynamics; 3) providing equations to determine the drift rate of the pressure and temperature sensors located inside the CRDS cavity from the CO2 and CH4 response function drift trends; 4) the determination of the H2O correction for CO has been performed using raw spectral peak data instead of the raw CO provided by the CRDS and using a running mean to smooth random noise in a long water-droplet test (12 hours) before

performing the least square fit; and 5) the existence of a small $H_2O$ dependence in the CRDS-flow and of a small spatial inhomogeneity in the temperature field inside the CRDS cavity are pointed out and their origin discussed.

The structure of this article is as follows. We firstly detail (Sect. 2) the results obtained in the initial tests performed on the Izaña CRDS (such as precision, repeatability, sensibility to inlet gas pressure and response function) as well as the

relationships between flow rate, CRDS inlet pressure and CRDS outlet valve aperture; and the pre-processing applied to the raw data (Sect. 3). Secondly, we analyse the results of the calibrations performed every 3-4 weeks since the end of 2015 using WMO tertiary standards, and provide some details on the response functions used and the numerical processing software developed to evaluate the calibrations (Sect. 4). Thirdly, some details of the obtained and implemented water vapour corrections are provided (Sect. 5). Finally, the ambient measurements carried out until July 2017 are presented, as

well as some details in the numerical processing software developed to obtain the ambient air CRDS measurements from raw data and calibration results, and compared with those obtained with other Izaña in situ measurement instruments (Sect. 6). In Sect. 7, a preliminary independent assessment on the drift rates of the CRDS CO standards is performed. The main conclusions are outlined in Sect. 8. A note concerning the inlet pressure (Sect. 2.4) and $H_2O$ dependences of the CRDS flow and the spatial inhomogeneity of the pressure field inside the CRDS cavity is presented (Appendix A). Additionally, we very

briefly describe a few more novelties in the Izaña GHG measurement programme since the WMO/IAEA Meeting on Carbon Dioxide, Other Greenhouse Gases and Related Tracers Measurement Techniques that took place in the year 2015 (GGMT-2015) in Appendix C.

## 2 Acceptance tests performed on the CRDS

A Picarro G2401 CRDS analyser (serial number CFKADS2196) for measuring $CO_2$, $CH_4$, CO and $H_2O$ was installed on

November 2015 at IZO. Several acceptance tests were performed on the CRDS at the station, roughly following the recommendations provided by the European Integrated Carbon Observation System (ICOS)-Atmospheric Thematic Centre (ATC) (ICOS-ATC, 2016). For processing the data associated with the tests, the raw values for $CO_2$ (not dry), $CH_4$ (not dry), CO and $H_2O$ of the "Synchronized DataLog_User" files were used.

### 2.1 Continuous measurement repeatability test

The first test was what ICOS-ATC (2016) called "Precision test". This test is called continuous measurement repeatability (CMR) by Yver Kwok et al. (2015), and consists in measuring a gas cylinder (filled with dry natural air) over 25 hours, rejecting the first hour as stabilization time. As Table 1 shows, the results (standard deviations) obtained in this test were well within the threshold established by ICOS-ATC (2016). Note that the precision for the $H_2O$ measurements indicated by the manufacturer of the CRDS is: <200 ppm for 5-second averages, and < 50 ppm for 5-minute averages. Therefore, it is

reasonable to obtain small (in absolute value) negative values (-2.8 ppm) for $H_2O$ when measuring dry air, since, taking into account the precision of the instrument, these values are completely compatible with 0.0 ppm.

| Raw data average length | CO2 S.D.(ppm) Obtained/ICOS-threshold | CH4 S.D.(ppb) Obtained/ICOS-threshold | CO S.D.(ppb) Obtained/ICOS-threshold | Mean H2O (ppm) |
|---|---|---|---|---|
| 1 minute | 0.013 / 0.050 | 0.19 / 1.0 | 0.87 / 2.0 | -2.8 |
| 60 minutes | 0.009 / 0.025 | 0.14 / 0.5 | 0.16 / 1.0 | -2.8 |

**Table 1. Results obtained during the "Precision test" (CMR) and ICOS-threshold values for two averaging times (1 and 60 minutes). The Standard Deviation (S.D.) reported is of the sample of 1-minute (or 60-minute) means obtained through the full duration of the test.**

### 2.2 Long-term repeatability test

The second test performed was a "Repeatability test" (this test is called long-term repeatability -LTR- by Yver Kwok et al., 2015). According to ICOS-ATC (2016), it consists in measuring alternately a gas cylinder (filled with dry natural air) during 30 minutes and ambient air (not dried) during 270 minutes over 72 hours. Statistics are based only on the last 10-minute-average data for each gas cylinder "injection". Indeed, we used 2 cylinders, each one measured every 5 hours, with the following measurement cycle: cylinder 1 during 40 minutes, wet ambient air during 20 minutes, cylinder 2 during 40

minutes, and wet ambient air during 200 minutes. Table 2 shows the results obtained for each tank and 10-minute average period: 20-30 minutes and 30-40 minutes. The results were well within the thresholds established by ICOS-ATC (2016). Note that the S.D. reported in Table 2 (associated to 10-minute means) are larger for CO2 and CH4 (but lower for CO) than the S.D.´s in Table 1 associated with 1-minute means. This means the decrease in the random noise due to the increase in the averaging time is countered by the increase in the S.D. due to the larger response drift in the longer test (this is not the case

for CO).

| Cylinder | 10-minute average period | CO2 S.D.(ppm) Obtained/ICOS-threshold | CH4 S.D.(ppb) Obtained/ICOS-threshold | CO S.D.(ppb) Obtained/ICOS-threshold | Mean H2O (ppm) |
|---|---|---|---|---|---|
| 1 | 20-30 min. | 0.016 / 0.050 | 0.23 / 0.5 | 0.23 / 1.0 | -1.3 |
| 1 | 30-40 min. | 0.021 / - | 0.34 / - | 0.28 / - | -1.8 |
| 2 | 20-30 min. | 0.016 / 0.050 | 0.23 / 0.5 | 0.35 / 1.0 | -1.2 |
| 2 | 30-40 min. | 0.026 / - | 0.31 / - | 0.31 / - | -1.4 |

**Table 2. Results obtained during the "Repeatability test" (LTR) and ICOS-threshold values. The Standard Deviation** (S.D.) reported is **of the sample of 10-minute means obtained through the full duration of the test.**

### 2.3 Ambient pressure sensitivity test

The third test performed was an ambient pressure sensitivity test for the measurements carried out during the mentioned period of 72 hours; i.e,, inlet gauge pressure was kept constant but the lab pressure (ambient pressure) was allowed to vary naturally. No pressure chamber was used, similarly to Yver Kwok et al. (2015). Note that this CRDS is not meant for use on

aircraft, where large ambient pressure changes might occur. This test provides an upper-limit for the ambient pressure sensitivity, since there might be instrumental drift not attributable to atmospheric pressure changes. The results obtained were: 0.04 ppb/hPa for CO, 0.0038 ppm/hPa for CO2, and 0.047 ppb/hPa for CH4. These sensitivities are not significant. The main purpose of this test was to confirm the CRDS CO measurements were not affected by natural ambient pressure

changes as indeed Yver Kwok et al. (2015) found for several CRDS units.

## 2.4 Relationships between inlet pressure, flow rate and outlet valve aperture: inlet pressure sensitivity test

Due to their importance for the present article, we detail firstly the relationships we have found between inlet pressure, flow rate and outlet valve opening.

According to the information provided by the manufacturer (Rella, private communication), there is both a proportional

valve and a physical critical orifice in the inlet system of the G2401 CRDS cavity. The proportional valve is opened slowly at start-up to ensure that the flow changes smoothly, but after this start-up procedure, the valve is set to open fully, and the flow is set by the critical orifice. There is also a proportional valve at the outlet of the cavity and upstream of the vacuum pump. The cavity (absolute) pressure is kept at 186.7 hPa (140 Torr) by controlling the opening of the Outlet Valve (OV).

A choked flow is a flow through a critical orifice in which the following condition holds for the inlet to outlet (absolute)

pressure ratio (Van den Bosch & Duijm, 2005):

$$\frac{p_i}{p_o} \geq \left(\frac{\gamma+1}{2}\right)^{\gamma/(\gamma-1)} \qquad (1),$$

where $p_i$ is the inlet (absolute) pressure, $p_o$ is the outlet (absolute) pressure, and $\gamma$ is the ratio of specific heats ($c_p/c_v$) for the considered gas. For dry air, $\gamma$ is equal to 1.4 and the right-hand side of Eq. (1) is equal to 1.893. In a choked flow, the speed is supersonic just downstream of the orifice and the flow rate does not depend on the downstream quantities (because

"information" cannot propagate faster than sound). Since the minimal recommended inlet (absolute) pressure for the CRDS G2401 is 400 hPa, and the cavity pressure is 187 hPa (140 Torr), the flow in the inlet critical orifice is always choked. This is the reason why the flow through the CRDS cavity depends only on upstream quantities (mainly the inlet pressure) and not on the cavity quantities. However, for flight-ready models, the critical orifice is located at the outlet of the CRDS cavity, and therefore, the inlet pressure for the orifice is the cavity pressure, which is kept constant.

The theoretical equation relating the standard volumetric flow ($F$) with the inlet quantities for a choked flow is (Van den Bosch & Duijm, 2005):

$$F = C_d \cdot A \cdot p_i \cdot \frac{T_s}{p_s} \cdot \left[\frac{\gamma \cdot R}{T_i} \cdot \left(\frac{2}{\gamma+1}\right)^{(\gamma+1)/(\gamma-1)}\right]^{1/2} \qquad (2),$$

where $T_s$ and $p_s$ are the standard (absolute) temperature (273.15 K) and pressure (1013.25 hPa), respectively, $T_i$ is the inlet (absolute) temperature, $R$ is the gas constant, $C_d$ is the discharge coefficient (dimensionless) and $A$ is the hole cross-sectional

area. Equation (2) shows that the standard volumetric flow is proportional to the inlet (absolute) pressure and inversely proportional to the square root of the inlet (absolute) temperature.

As indicated in Sect. 6, the ambient air/gas standard plumbing configuration operational since 28 November 2016, includes a Red-Y mass flow meter (MFM) downstream of the CRDS, but upstream of the vacuum pump. There is an expansion volume between the MFM and the vacuum pump to smooth the pulses induced in the flow by the pump.

The fourth acceptance test was an inlet pressure sensitivity test when measuring a gas cylinder filled with dry natural air. This test was performed on 25 and 26 November 2015. Additional more complete tests of that type were performed in July and August 2018.

Figure 1 shows results of that second set of tests (no flow rate measurements were carried out during the first set of tests) performed on 4 non-consecutive days in which dry natural air coming from a standard was measured during many consecutive hours changing the CRDS inlet pressure every hour or longer, discarding the first 20 minutes after the pressure change, and keeping the mean of the measurements till the next pressure change. The sequences of pressures used were not monotonic but with jumps up and down. Pressure was measured using the regulator gauge and then ambient pressure was summed. On 24 July 2018 the test was performed using a cylinder regulator of a different type than the one used the other 3 days. As Fig. 1 shows, the relationship between the CRDS inlet pressure and the flow rate is linear. Table 3 shows the coefficients and $R^2$ obtained fitting a straight line to the data of each test. The linear fits are very good ($R^2$ is almost 1) and very similar (the first one is slightly different probably due to the use of a different regulator and a larger pressure range).

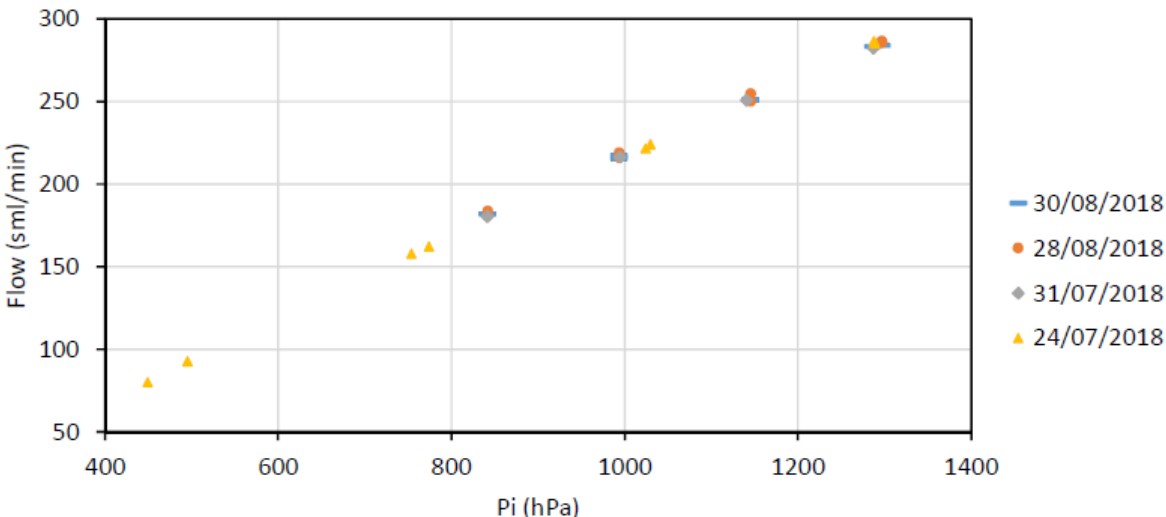

**Figure 1. Results of tests performed on 4 non-consecutive days in which dry natural air coming from a standard was measured during many consecutive hours changing the CRDS inlet pressure every hour or longer. Pi is the CRDS inlet pressure and Flow is the flow measured downstream of the CRDS.**

| Test | Slope (sml/[min·hPa]) | Intercept (sml/min) | $R^2$ |
|---|---|---|---|
| 24/07/2018 | 0.244 | -27.9 | 0.9998 |

| | | | |
|---|---|---|---|
| 31/07/2018 | 0.229 | -12.2 | 0.9996 |
| 28/08/2018 | 0.228 | -9.5 | 0.9984 |
| 30/08/2018 | 0.226 | -8.4 | 0.9996 |

**Table 3. Coefficients and $R^2$ obtained fitting a straight line to the test data shown in Fig. 1.**

Figure 2 shows results of the first and second sets of inlet pressure sensitivity tests. The first tests were performed alternating only between two inlet pressures, using cylinder regulators different from those used in the second set of tests. As Fig. 2 shows, the relationship between the CRDS inlet pressure and OV is linear. Table 4 shows the coefficients and $R^2$ obtained fitting a straight line to the data of each test. The linear fits are very good ($R^2$ is very near 1) and have very similar slopes (except for the test performed using a larger pressure range).

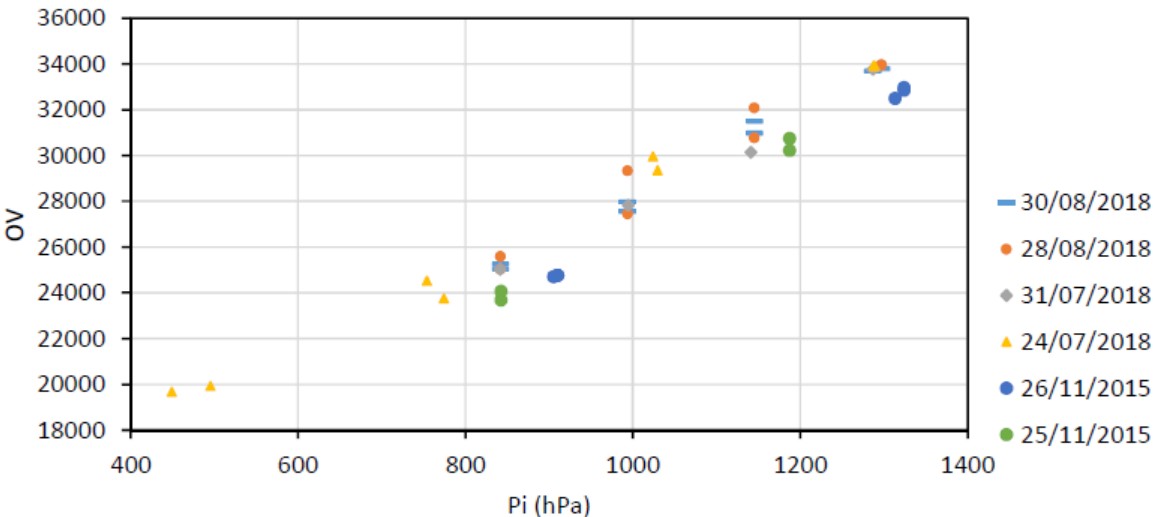

**Figure 2. Results of the first and second sets of inlet pressure sensitivity tests. Pi is the CRDS inlet pressure and OV is the aperture of the CRDS-cavity outlet valve.**

| Test | Slope (hPa$^{-1}$) | Intercept | $R^2$ |
|---|---|---|---|
| 25/11/2015 | 19.2 | 7720 | 0.9954 |
| 26/11/2015 | 19.5 | 7020 | 0.9996 |
| 24/07/2018 | 17.6 | 11300 | 0.9913 |
| 31/07/2018 | 19.2 | 8710 | 0.9913 |
| 28/08/2018 | 19.1 | 9320 | 0.9659 |
| 30/08/2018 | 19.5 | 8660 | 0.9934 |

**Table 4. Coefficients and $R^2$ obtained fitting a straight line to the test data shown in Fig. 2.**

Figure 3 shows results of the inlet pressure sensitivity tests for CO2, CH4 and CO, using OV as independent variable. Since the cylinders used were not unique, for each test we use the difference with respect to the mean raw mole fraction measured during the whole test and the symbol "d" to indicate such difference. As Fig. 3 shows, the relationship between dCO2raw and OV is linear. Note that the dCO2raw values are quite small and therefore it is not surprising there is some noise in the fit, but the slopes obtained in the 6 tests are consistent. For dCH4raw there is a linear relationship too, but noisier. For dCOraw there is not any significant linear relationship. Table 5 shows the mean slope and intercept for each trace gas as well as the associated standard deviations. Those numbers confirm the previous sentences concerning the statistical significance of the linear fits for CO2, CH4 and CO.

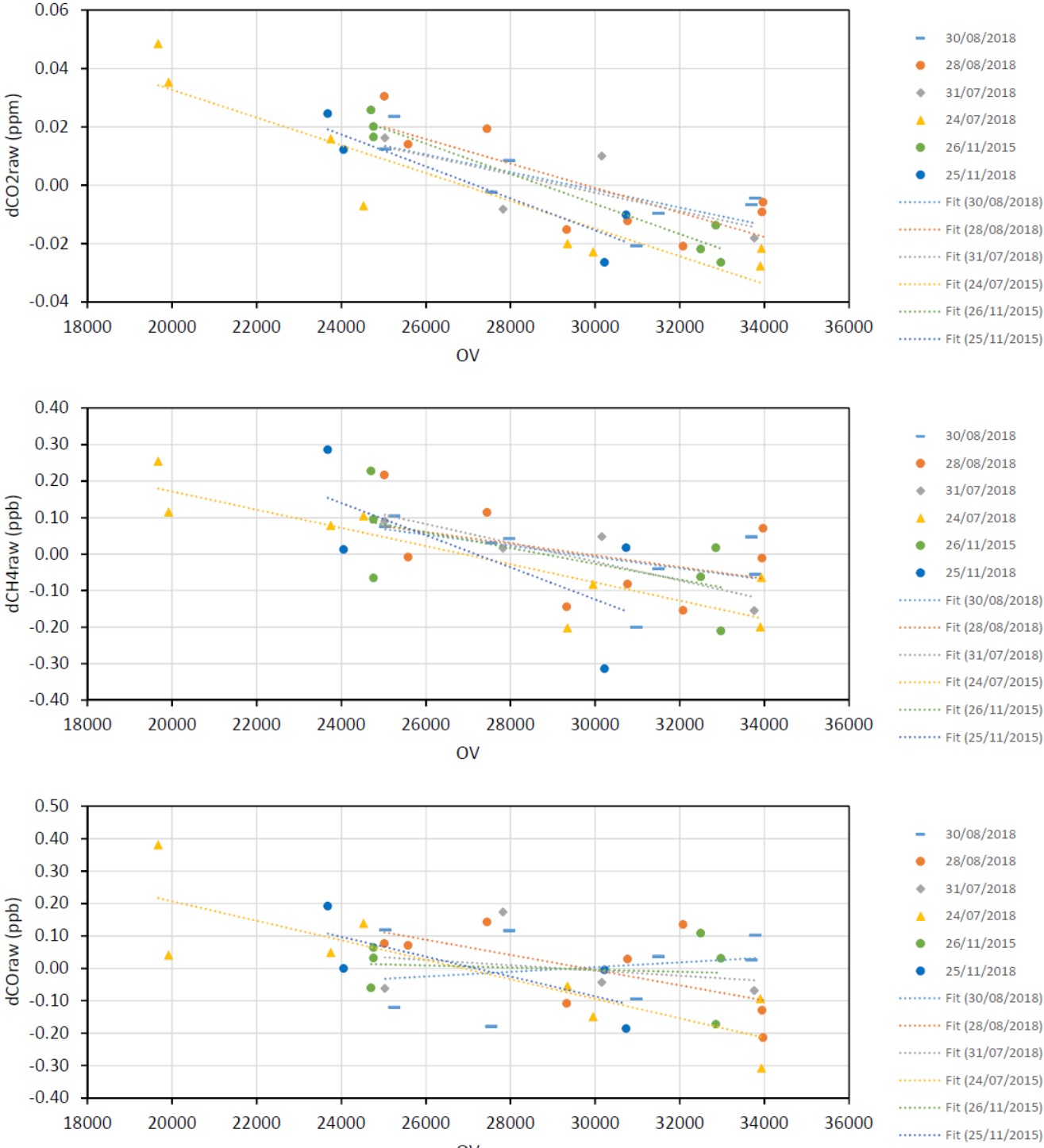

**Figure 3. Results of the inlet pressure sensitivity tests for CO2, CH4 and CO, using OV as independent variable. For each test, "d" denotes the difference with respect to the mean raw mole fraction measured during the whole test.**

| Trace gas | Mean slope | Stand.Dev. | Mean intercept | Stand.Dev. |
|-----------|------------|------------|----------------|------------|
| CO2 | $-4.29 \cdot 10^{-6}$ ppm | $1.03 \cdot 10^{-6}$ ppm | 0.122 ppm | 0.026 ppm |
| CH4 | $-2.46 \cdot 10^{-5}$ ppb | $1.05 \cdot 10^{-5}$ ppb | 0.695 ppb | 0.271 ppb |
| CO | $-1.47 \cdot 10^{-5}$ ppb | $1.56 \cdot 10^{-5}$ ppb | 0.409 ppb | 0.433 ppb |

**Table 5. Mean slope and intercept for the fits (Fig. 3) associated with each trace gas (as well as the associated standard deviations) in the inlet pressure sensitivity tests.**

Taking into account the linear relationship between inlet pressure and OV, and the values shown in Table 5, the sensitivities obtained were: $0.00 \cdot 10^{-2}$ ppb/kPa (since the slope is not statistically significant) for CO, $-0.83 \cdot 10^{-3}$ ppm/kPa for CO2, and $-0.47 \cdot 10^{-2}$ ppb/kPa for CH4, which are quite small except for CO2 (e.g., 30 kPa produces a bias of -0.025 ppm in CO2). Since the CRDS inlet pressure can be different (e.g., differences of a few tens of kPa may be present) when changing the sample (laboratory standards, target gases and ambient air), and in order to be able to achieve a very accurate response function for CO2, we empirically correct for this effect as explained in Sect. 3.1.

### 2.5 Calibration curve fitting test

The fifth test was fitting response curves when calibrating with four WMO tertiary standards (6 cycles, 2.5 hours/cycle, each standard is measured during 30 consecutive minutes as well as a target gas). When performing a linear fitting for CH4, the Root Mean Square (RMS, accounting for the effective degrees of freedom) residual was 0.143 ppb (very small). When performing a linear fitting for CO, the RMS residual was 0.067 ppb (very small). However, when performing a linear fitting for CO2, the RMS residual was 0.0395 ppb, which is larger than the values we usually obtain with our NDIR-based measurement system for CO2 when using a quadratic fitting (Gomez-Pelaez & Ramos, 2011). Moreover, when using a quadratic fitting for the CRDS CO2, the RMS residual was 0.0284 ppm, still slightly worse than that of our NDIR-based systems. Finally, when correcting the raw CO2 from the inlet pressure sensitivity and then performing a quadratic fitting, the RMS residual was 0.0219 ppm, which is similar to the values obtained with our NDIR-based systems. During the calibration described above, the inlet pressure differences were especially intense. We have improved our skills since then getting smaller inlet pressure differences during the calibrations.

### 3 Data acquisition and pre-processing

Pre-processing refers to the computation of raw-data 30-second means and corresponding standard deviations using the DataLog_User files (not the synchronized ones), as well as the computation of some derived variables detailed below. We

have developed the pre-processing software, as well as the calibration and ambient-air processing software. For the computation of 30-second means and standard deviations, we take into account the so called "species" field. The species code is different for each spectral range in which the measurements are performed. There are 4 spectral ranges and 3 lasers, since two nearby spectral ranges are scanned with the same laser (see Chen et al., 2010, and Chen et al., 2013, for details about these spectral ranges). The fields updated in each file line are those related with the spectral lines measured in the corresponding spectral range. The DataLog_User file contains 1.7 lines/second and these lines follow this time sequence of species: 1, 2, 4 and 3. The raw variables associated with each "species" value are: a) Species=1, CO2, peak_14 ($p14$), and CO; b) Species=2, CH4, CO2_dry; c) Species=3, H2O, h2o_reported ($hr$), CO2_dry, and CH4_dry; and d) Species=4, CO, b_h2o_pct ($bh$), and peak84_raw ($p84$); where CO2 and CH4 are raw values not corrected from H2O dilution nor pressure broadening, whereas CO2_dry and CH4_dry have been (factory) corrected from those effects, $p14$ is the raw value associated with the main CO2 peak, CO is already (factory) corrected from the CO2 and H2O influences, H2O is the calibrated value obtained from $hr$, which is the reported H2O raw value associated with the main H2O peak (in the CH4-peak laser wavelength range), $bh$ is the H2O raw value associated with the secondary H2O peak (in the CO-peak laser wavelength range), and $p84$ is the raw value associated with the CO peak. Additionally, there are other raw variables not associated with a single "species" value that we use: Cavity Pressure ($CP$), Cavity Temperature ($CT$), Multi Position Valve (MPV) Position ($MPVP$), Outlet Valve ($OV$), and Solenoid Valves ($SV$). Note that the Picarro software is able to control both solenoid (up to 6) and MPV (just one) valves for the gas handling system, and stores the positions of the valves in the same file than the output data for the measured trace gases. The information on the positions of the solenoid valves is codified in a single natural number (SV).

A 30-second mean is accepted when: a) at least 85% of the expected data is present; and b) all the instantaneous data have the same $MPVP$ and $SV$ values. Additionally, a counter called $npcmc$ is assigned to each 30-second mean. It indicates the number of consecutive 30-second means with the same configuration for both, $MPVP$ and $SV$.

### 3.1 Inlet pressure sensitivity correction for raw CO2

We use the original empirical relationship we determined through the inlet pressure sensitivity test mentioned in Sect. 2.4, which leads to the following Outlet-Valve-corrected raw CO2:

$$CO2ovc = CO2raw + 0.04 \cdot (OV - 26468.15)/7700 \qquad (3).$$

CO2ovc is the raw value we use for the CO2 processing. These slope and intercept values are compatible with the corresponding mean values shown in Table 5 (distant less than 1 standard deviation from the corresponding mean). CO2ovc is corrected from the inlet pressure sensitivity but not for the H2O dilution and pressure broadening. According to the knowledge of the authors, this is a new correction not considered previously in the scientific literature.

Note that if Eq. (3) is expanded, only the slope term in OV is kept fixed, since the independent term is, in practice, combined with the independent term of the mole-fraction calibration curve (see Sect. 4), which is updated periodically. Therefore, this automatically takes into account hypothetical drifts in the independent term of Eq. (3). As Fig. 2 and Table 4 show, the slope

of the linear relationship between OV and Pi is consistent through the years, and no significant change in time is observed in the slope of the linear relationship between dCO2raw and OV. Note that if we average the addends of Eq. (3) and subtract those to Eq. (3), we obtain an equation for dCO2raw that is linear in OV and has an intercept that depends on the average OV. This explains why in Fig. 3a, the fit for the test with the largest OV range (smallest average OV) is below the rest of test fits.

In Appendix A, we provide a plausible physical explanation for this effect, after determining the H2O dependence of the CRDS flow and arguing that the pressure field inside the CRDS cavity is slightly inhomogeneous. Note that our arguments that point out to the fact that the inhomogeneity of the pressure field inside the cavity due to the flow produces the CO2raw dependence on OV, seem to be also supported by the following fact: the relative effect of the flow on CO2raw and CH4raw is roughly the same, as we could roughly expect from the decrease in trace gas concentration (mol per volume) that takes place inside the CRDS cavity near the inlet and outlet, due to the decreased pressure (see Appendix A). Note that dividing the slopes of Table 5 by the typical ambient air mole fraction at IZO (405 ppm for CO2 and 1880 ppb for CH4) we obtain: $1.06 \cdot 10^{-8}$ for CO2 and $1.31 \cdot 10^{-8}$ for CH4.

## 3.2 Raw CH4

We call CH4raw the raw CH4 multiplied by 1000 (to have ppb units) and use it for the CH4 processing. Note that CH4raw is the wet value not corrected from H2O dilution and pressure broadening.

## 3.3 Computed raw wet CO

The CO value provided by the G2401 CRDS includes the correction due to H2O dilution and pressure broadening. However, what we use for CO processing is the CO raw value (wet) that Picarro calls peak84_spec_wet (*p84sw*), which is *p84* corrected from the H2O and CO2 spectral peak overlapping (interference that changes the zero of *p84*). To compute it, we use the equations that our G2401 employs internally (Rella, private communication):

$$co2\_p14 \ = \ 0.706630873 \cdot p14 \tag{4},$$

$$p84sw \ = \ p84 + off + w1 \cdot bh + wc \cdot bh \cdot co2\_p14 + w2 \cdot bh^2 + c1 \cdot co2\_p14 \tag{5},$$

where *off*=-0.000800885106752, *w1*=-0.0334069906515, *wc*=-8.2480775807e-7, *w2*=0.00633381386844, and c1=8.87510231866e-6. We denote *p84sw', p84sw* multiplied by 1000. That is the raw value we use for the CO processing, i.e., it is the wet value not corrected from H2O dilution and pressure broadening nor converted to ppb units.

## 4 Calibrations and Response Functions

After processing, the measurements we carry out with the CRDS are in the following WMO scales: X2007 for CO2, X2004A for CH4 and X2014A for CO, since we use four multi-species WMO tertiary standards filled (with dried natural air) and calibrated by the WMO Central Calibration Laboratory (CCL) for these gases (https://www.esrl.noaa.gov/gmd/ccl/).

In each cycle of a calibration we use the four WMO tertiary standards and two target gases that act as unknowns. Each tank is measured continuously during 30 minutes every cycle. From December 2015 to August 2016, each calibration had 5 cycles and a calibration was performed every 3 weeks. From September 2016 till present, each calibration has 2 cycles and a calibration is performed every month. In the first period, we adopted a conservative strategy and after analysing the obtained results in detail we concluded that the second calibration strategy provided results that satisfied our accuracy requirements. Note that since there are technicians at the station every day, the regulators of the WMO tertiary standards remain closed between calibrations. This helps avoid any hypothetical problem of drifting in the standards due to very small leaks or differential diffusion inside the regulators that might propagate to the interior of the cylinders through the open valves by diffusion during the weeks the standard air remains static inside the regulators. For CO2 and CH4, the last 10 minutes of each gas injection are used. However, for CO, the last 20 minutes are used since CO measurements are noisier (i.e., better signal to noise ratio when incrementing the averaging period) and 10 minutes of stabilization time is enough for CO (numerical details not presented here).

The calibration processing is done using our own numerical code. For processing a calibration, the code computes the mean raw response for each tank and species, and then performs a least-square fit to the respective response function detailed below. For CH4 and CO, we use linear response functions:

$$CH4raw = b \cdot CH4 + c \qquad (6), \text{ and}$$

$$p84sw' = b \cdot CO + c \qquad (7),$$

where CH4 and CO are the dry mole fractions (the gas standards are dry) in ppb assigned by the CCL on the WMO scales. We have preferred to use a quadratic fit instead of a linear fit for CO2, since the RMS residual is significantly smaller: considering the first 13 calibrations, the mean RMS residual for linear fits is 0.035 ppm, whereas for quadratic fits it is 0.020 ppm. As mentioned in Sect. 2.5, we have chosen the quadratic fit for CO2 to obtain RMS residuals as smaller as we obtain with the IZO NDIR-based measurement system. The quadratic function with raw signal slightly corrected in the outlet valve aperture used (as described in Sect. 3.1) is:

$$CO2ovc = a \cdot CO2^2 + b \cdot CO2 + c \qquad (8),$$

where CO2 is the dry mole fraction in ppm assigned by the CCL on the WMO scale. Since $b$ is always positive (and near 1) and $a$ is always negative and near zero, CO2 is given by this solution of the second-order algebraic Eq. (8):

$$CO2 = \left[ -b + \sqrt{b^2 - 4 \cdot a(c - CO2ovc)} \right] / (2 \cdot a) \qquad (9).$$

To assess the drift in time of the response function from December 2015 to July 2017, we use the concept of virtual tank of fixed (assigned WMO; i.e., *CO2, CH4,* and *CO*) mole fractions (Yver Kwok et al., 2015) and compute the raw values (*CO2ovc, CH4raw,* and *p84sw'*) associated with those mole fractions using the response functions obtained in the calibrations. We consider a virtual tank with 400 ppm of CO2, 1850 ppb of CH4 and 100 ppb of CO. Moreover, we present here a justification of that procedure and complement it by using the local slope of the response function at the mole fraction of the virtual tank for each species. In the field of high-accuracy atmospheric trace-gas measurements, the calibration fits are generally performed using a limited range in the independent variable that is far from zero (a range around the atmospheric

mole fractions of interest). This produces an anticorrelation between the coefficients $b$ and $c$. The reason is the fact that if $b$ has a positive error (larger slope) then $c$ will have a negative error (smaller Y-intercept) that will increase in absolute value as the distance between the used X-range and zero increases. Therefore, plotting the time series $b$ and $c$ is not the best option for assessing the stability in time of the response function, since part of the variability is due to the anticorrelation and does not correspond with a real variability within the X-range of interest. A more interesting option is to plot the Y-value corresponding to a X-value located within the range of interest instead of $c$ (virtual tank concept) and the local slope at that X-value. Note that for CH4 and CO, which have linear response functions, the local slope does not depend on mole fraction and is equal to the coefficient $b$.

Figure 4 shows the CRDS raw responses for that virtual tank computed using the response functions obtained in the calibrations. From Figure 4, we obtain the CRDS long-term drift of the raw responses: 0.104 ppm/year for CO2, 2.22 ppb/year for CH4, and 0.544 ppb/year for CO.

As Yver Kwok et al. (2015), we define the fractional variables CH4frac=CH4raw/1850 and CO2frac=CO2ovc/400, i.e., as the raw mole fractions of the virtual tank divided by the real mole fractions. Figure 5 shows a scatter plot of CH4raw versus CO2ovc for the virtual tank. When fitting a linear function, the obtained slope is 20.788 ppb/ppm, which is equal to 4.495 when using the fractional variables, CH4frac and CO2frac.

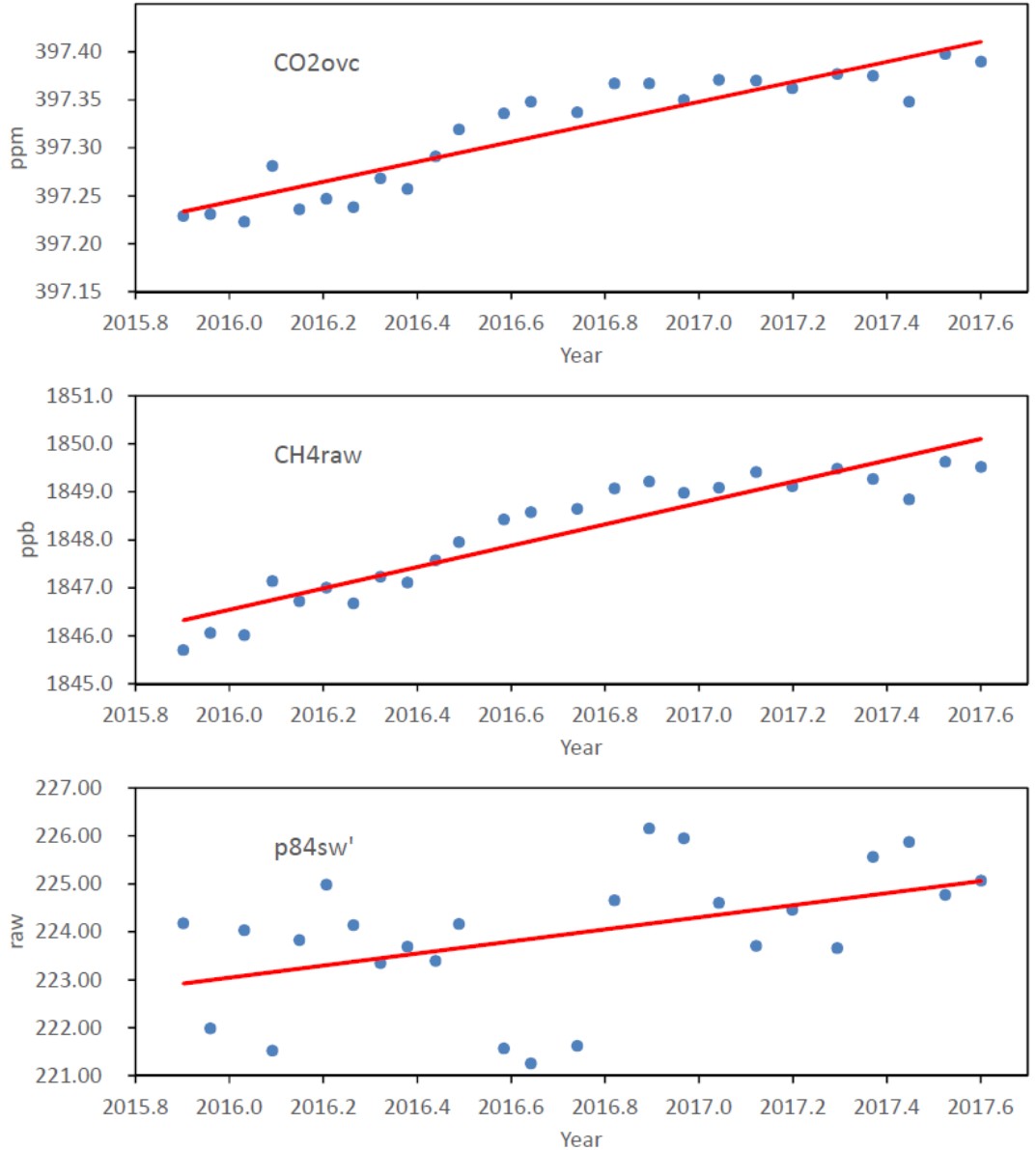

**Figure 4: CRDS raw responses (blue dots) for a virtual tank (400 ppm of CO2, 1850 ppb of CH4, and 100 ppb of CO) computed using the response functions obtained in the calibrations. Each red line corresponds to a least-square fitting of the data to a linear function.**

Assuming that those drifts are due to drifts in the real pressure and temperature of the cavity (i.e., drifts in the P and T sensors that, in turn, cause the cavity to be controlled at slightly drifting pressure and/or temperature), taking into account the

fact that the empirical sensitivities (partial derivatives) of the CRDS raw CO2 and CH4 with respect to *CP* and *CT* are known (Sect. 3.3.6 of Yver Kwok et al., 2015), and using these general relationships between partial derivatives:

$$\frac{dCH4frac}{dCO2frac} = \left(\frac{\partial CH4frac}{\partial T} \cdot \frac{dT}{dp} + \frac{\partial CH4frac}{\partial p}\right) / \left(\frac{\partial CO2frac}{\partial T} \cdot \frac{dT}{dp} + \frac{\partial CO2frac}{\partial p}\right) \quad (10), \text{ and}$$

$$\frac{dCH4frac}{dt} = \left(\frac{\partial CH4frac}{\partial T} \cdot \frac{dT}{dp} + \frac{\partial CH4frac}{\partial p}\right) \frac{dp}{dt} \quad (11).$$

We have determined that our CRDS has a long-term drift of 0.114 °C/hPa (obtained using Eq. 10 and taking into account that the left-hand side of Eq. 10 is equal to the slope shown in Fig. 5 multiplied by 400/1850), 0.595 hPa/year (obtained using Eq. 11 and taking into account that the left-hand side of Eq. 11 is equal to the slope shown in Fig. 4b divided by 1850), and 0.0678 °C/year (obtained multiplying the two former numbers) in the cavity sensors. Note that the use of Eqs. (10) and (11), which provide a quantitative estimation of the drift in temperature and pressure, is new in the GHGs monitoring literature.

The fact that the drifts in temperature and pressure are linear in time is a consequence of the fact that the slopes shown in Fig. 5, Fig. 4b and Fig. 4a are constant, and from Eqs. (10) and (11).

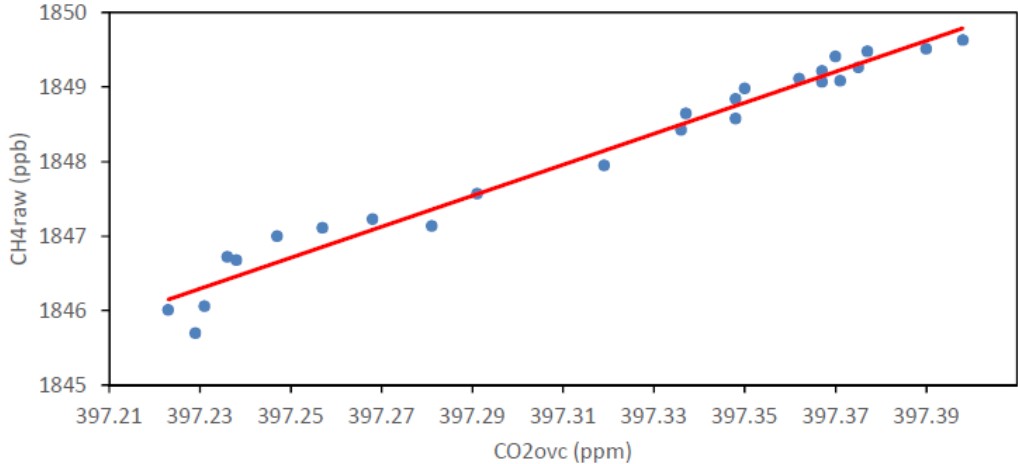

**Figure 5: Scatter plot of CH4raw versus CO2raw for the mentioned virtual tank. The red line has been obtained performing a least-square fitting of the data to a linear function.**

Figure 6 provides additional information about the response functions determined in the calibration. In detail, Fig. 6 provides for each species the local slope of the response function at the mole fraction of the virtual tank, as well as the quadratic coefficient (*a*) for the CO2 response function.

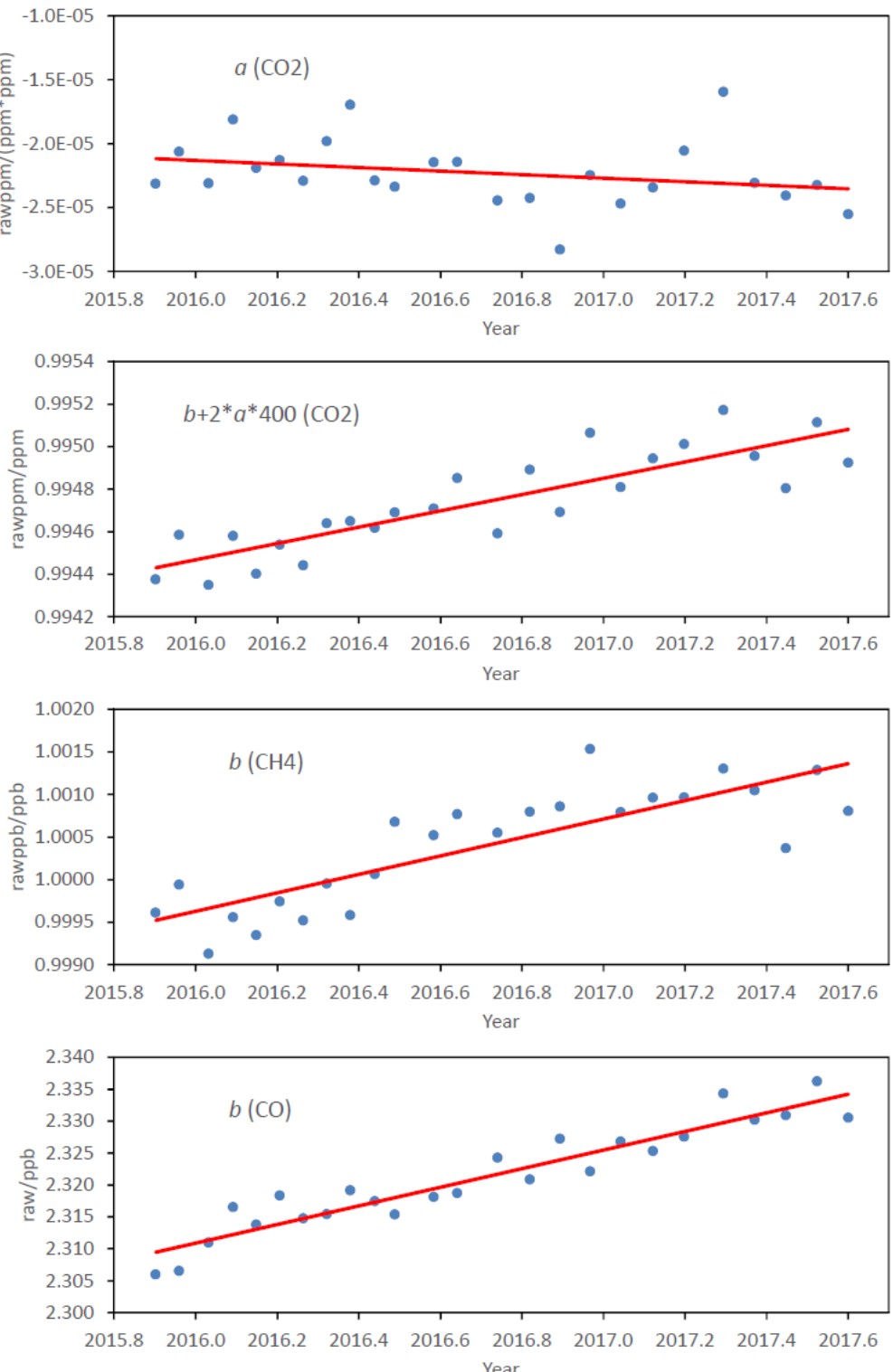

**Figure 6: Quadratic coefficient of the CO2 response function (*a*) and local slope of the response function at the mole fraction of the virtual tank, for CO2, CH4 and CO. Note that for CH4 and CO, the local slope does not depend on mole fraction and is equal to the coefficient *b*. The red line corresponds to a least-square fitting of the data to a linear function.**

Figure 7 provides for each species and calibration the RMS residual of the fit and the difference between the assigned mole
fraction to a target gas and the mean mole fraction of such target gas. For CO2, the mean RMS residual for all the
calibrations is 0.021 ppm, there is no trend in the associated time series, and the maximum departure in absolute value of a
target gas assignment from the mean for such target gas is 0.026 ppm. Those numbers are quite small compared with the
GAW DQO for CO2, and indicate a good performance of the measurement system for CO2. For CH4, the mean RMS
residual for all the calibrations is 0.09 ppb, there is no trend in the associated time series, and the maximum departure in
absolute value of a target gas assignment from the mean for such target gas is 0.18 ppb. As for CO2, those numbers are quite
small compared with the GAW DQO for CH4 and indicate a good performance of the measurement system for CH4.
However, for CO, there is a significant trend in the time series of RMS residuals, which show a value of around 0.1 ppb for
the first calibrations whereas increasing to around 0.7 ppb for the last calibrations. Moreover, there are significant downward
drifts in all the time series of target gas assignments, which is a very strange fact since CO standards normally drift upward
(i.e., when a CO standard drifts, this drift is generally positive; e.g., https://www.esrl.noaa.gov/gmd/ccl/co_scale.html). All
these facts suggest that the IZO CRDS CO WMO tertiary standards might be drifting upward quite significantly at different
rates. In Sect. 7, this possibility is investigated.

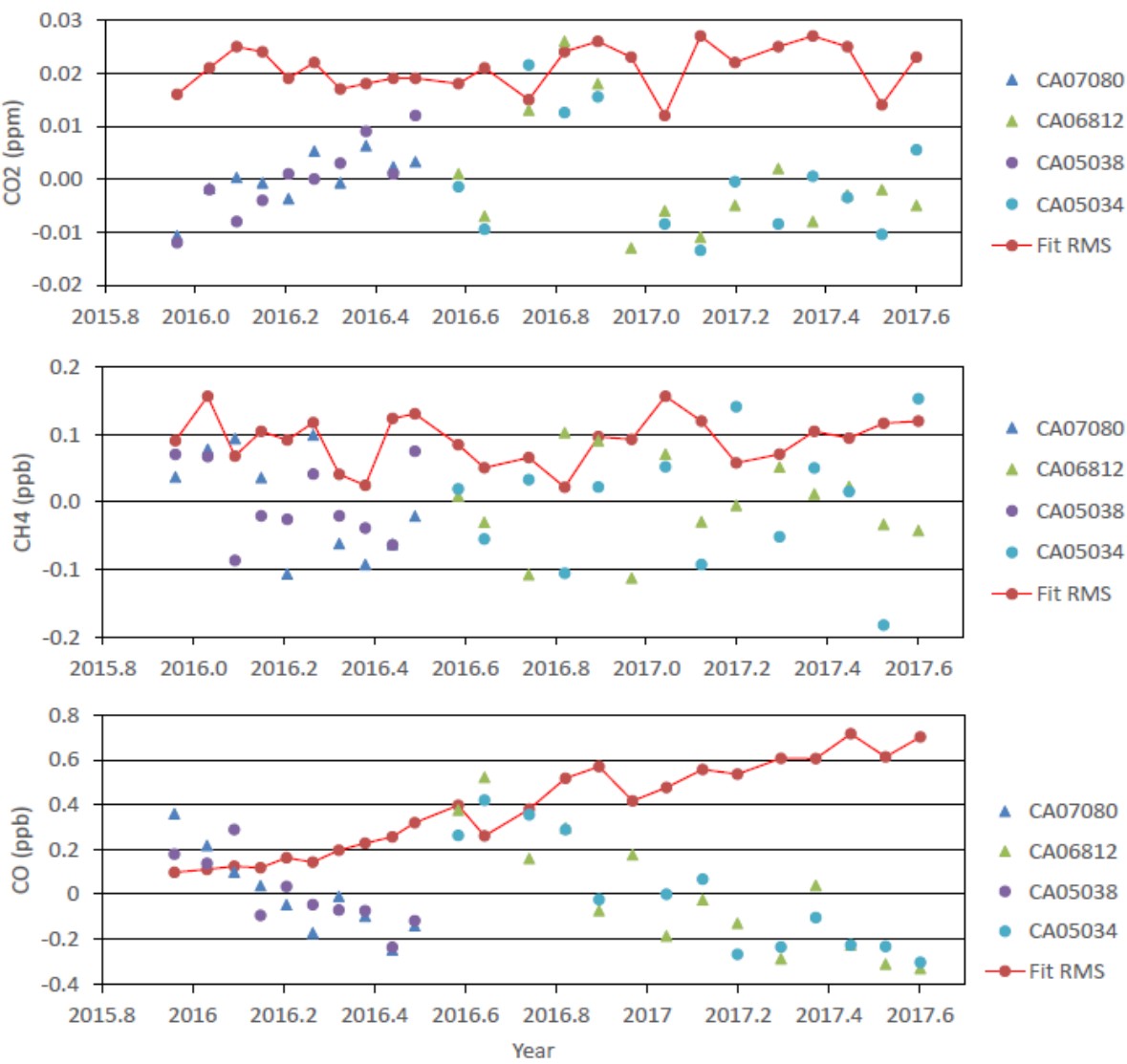

**Figure 7: RMS residual of the fit (red dots/line), and difference between the assigned mole fraction to a target gas and the mean mole fraction of such target gas (a different colour is used for each target gas), for each species and mole-fraction calibration performed using four WMO laboratory standards (every 3 or 4 weeks).**

5 **5 Water Vapour Correction: water droplet method**

The natural air contained in the WMO tertiary standards and the target standards is dry. However, ambient air contains water vapour and if it is not completely dried before measurements (as is done in NDIRs), the dilution and pressure broadening

effects due to H2O need to be taken into account and corrected (Chen et al., 2010; Zellweger et al., 2012; Rella et al., 2013, Chen et al., 2013; Karion et al., 2013).

For determining the particular water vapour dilution and pressure broadening corrections for the IZO CRDS G2041, we performed a long water droplet test (around 12 hours) using crushed (to increase the surface/volume ratio) Silica Gel balls soaked with deionized water contained in a stainless steel filter housing (called MPI/NOAA implementation in Rella et al., 2013), as Fig. 8 shows. The dry natural air coming from a standard flowed continuously during around 12 hours through the wet Silica Gel before being measured in the CRDS. Figure 9 shows the evolution in time of the h2o_reported (*hr*) in pph (parts per hundred in mole fraction; i.e., centimoles per mole of air) determined by the CRDS.

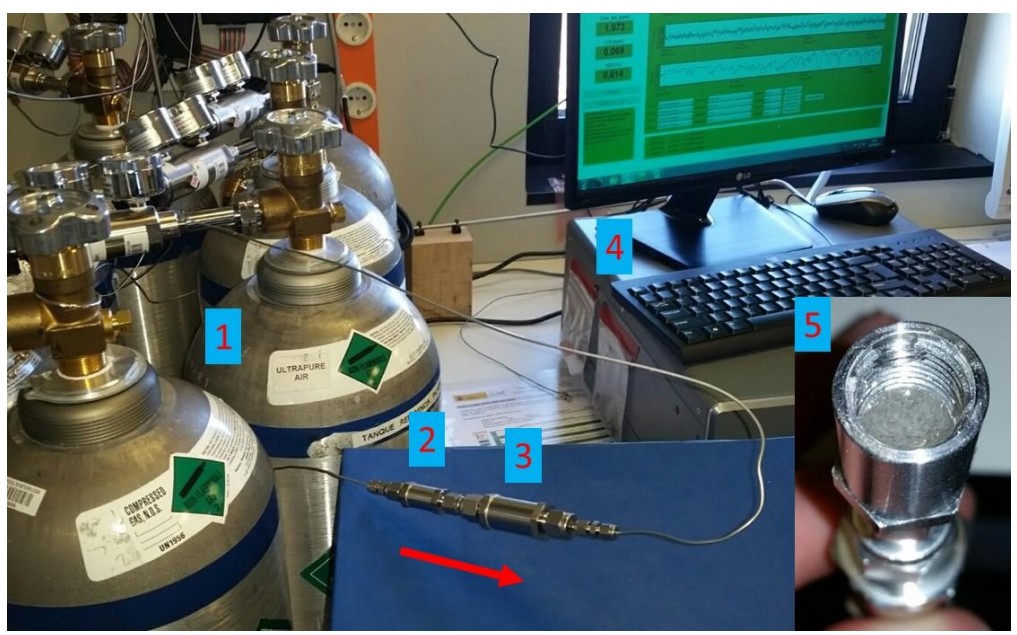

**Figure 8: Experimental setup used for the water droplet test performed on the IZO CRDS G2401. 1) Cylinders containing dry natural air (the red arrow shows the flow direction); 2) stainless steel filter housing containing crushed Silica Gel balls (see details of the interior in 5) soaked with deionized water; 3) stainless steel 0.5-µm filter; and 4) CRDS.**

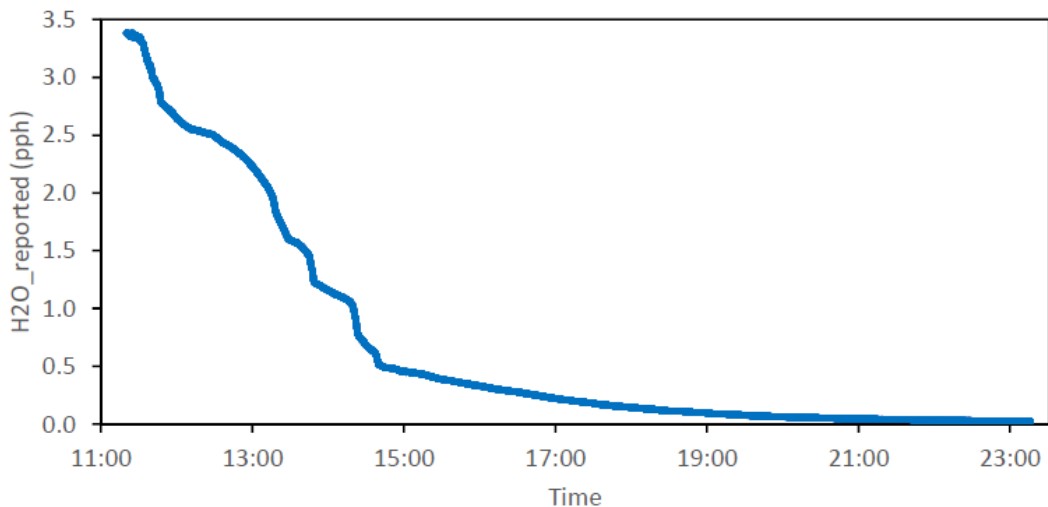

**Figure 9: Evolution in time of the h2o_reported (pph) determined by the CRDS.**

For CO2 and CH4, we use these empirical correction equations for the H2O dilution and pressure broadening effects (Chen
et al., 2010):

$$CO2ovc\_wet = CO2ovc\_dry \cdot (1 + d \cdot hr + e \cdot hr^2) \qquad (12),$$

$$CH4raw\_wet = CH4raw\_dry \cdot (1 + d \cdot hr + e \cdot hr^2) \qquad (13),$$

where CO2ovc_dry, CH4raw_dry, $d$ and $e$ (different in each equation) are determined by least-square fitting to test results.
Since the experiment is very long, before performing the least-square fit, we aggregate data computing 100-data means. That
corresponds approximately to a 59-second mean, since there are 1.7 data per second. For CO2, we obtained coefficients $d$
and $e$ very close to those reported by Chen et al. (2010). Thus, we decided to use their coefficients, $d$ = -0.012 pph[-1] and $e$ = -
0.000267 pph[-2]. However, in Appendix B, we provide Figure B1, which shows CO2ovc_wet versus $hr$ during the
experiment, because it is going to be referenced in the present section when discussing a related topic. Figure 10 shows
CH4raw_wet versus $hr$ during the experiment and the least-square fitted curve, the coefficients obtained for CH4 being: $d$ = -
0.009974 pph[-1] and $e$ = -0.0001757 pph[-2].

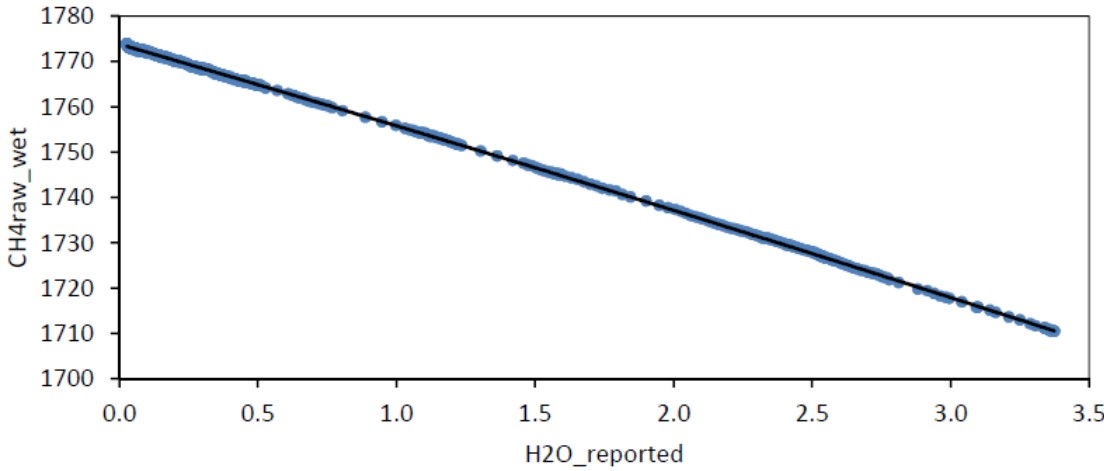

**Figure 10: CH4raw_wet versus *hr* during the water droplet experiment (in blue) and the least-square fitted curve (in black).**

As requested by one of the referees, we discuss here the reported sensitivity of the CRDS pressure sensor with H2O by
Reum et al. (2019). First at all, we point out we have found the fits of the standard H2O correction models (Eqs. 12 and 13)
to our empirical data are excellent (see Figs. 10 and B1). Note we have used a long water droplet test instead of the common
short droplet test used by Reum et al. (2019). Those authors also used a method that maintains stable water vapor levels, but
only considers a discrete set of levels. Since they also found a large disparity of results between different units of the same
CRDS model, we recommend further investigation on the pressure sensor sensitivity claimed by Reum et al. (2019) taking
into account the following two facts. First, when attaching the external pressure measurement unit (Fig. 2 of Reum et al.,
2019), the choked flow takes place at the needle valve called "choke" instead of at the CRDS outletvalve. However, at the
CRDS outletvalve, which has a critical orifice in the flight-ready CRDS used by those authors, there is a significant
longitudinal decrease of the pressure due to viscosity. Note that if the cavity pressure (CP) is increased, the flow rate
increases (see our Eq. 2, but using CP instead of $p_i$) and the pressure drop due to viscosity increases. This seems to be indeed
the reason that explains the 0.95 value those authors found for the derivative of the external pressure with respect to the
CRDS pressure. Note also that humid air has a larger dynamical viscosity than dry air (e.g., Tsilingiris, 2018). For H2O mole
fraction between 0 and 5 pph, it holds that dynamical viscosity depends almost linearly on H2O mole fraction, being equal at
45 ℃ to $1.94 \cdot 10^{-5}$ N$\cdot s \cdot m^{-2}$ for 0 pph, whereas it is equal to $2.04 \cdot 10^{-5}$ N$\cdot s \cdot m^{-2}$ for 5 pph (obtained from the Tsilingiris,
2018, equations). This fact agrees in sign with the 1% decrease reported by Reum et al. (2019) on the derivative of the
external pressure with respect to the CRDS pressure when considering humid air (3 pph). Second, on the tee to which the
external pressure sensor is attached, there is a H2O gradient between the main flow and the dryer, and a net H2O diffusive
flux from the main flow till the dryer, which might produce a drag force in the dry air. Then, this dry air would need to adopt
a configuration with a pressure gradient force opposing to the H2O drag force. This would mean the external pressure sensor

would be measuring a pressure larger than that presents in the main flow, and the pressure difference would increase with H2O mole fraction.

We consider here briefly, in order to suggest future research on this topic, a fact that has not been taken into account in the previous CRDS literature according to our knowledge. Due to the CRDS inlet critical orifice, at the same time as the velocity increases until it becomes supersonic, there is an adiabatic expansion of the air that decreases the temperature below the dew/frost point if the air has enough water vapour. Note that at the critical orifice the velocity becomes sonic, the pressure is 0.528 times de inlet pressure, the density is 0.634 times the inlet density, and the absolute temperature is 0.833 the inlet absolute temperature (Courant and Friedrichs, 1976). Those values decrease more downstream of the critical orifice till the air reaches the stationary shock front, where the velocity decreases suddenly and the temperature increases suddenly until reaching a value similar to the inlet one (Courant and Friedrichs, 1976). We suggest future research on the following possibility: formation of micro-droplets in the expansion flow, dissolution of CO2 in these micro-droplets, and interchange of oxygen atoms between the dissolved CO2 and the liquid water (therefore, the isotopic ratios are changed), before the micro-droplets are suddenly evaporated at the shock front.

For CO, we rely on the H2O dilution and pressure broadening correction determined by Rella (2010), as other authors have done (Chen et al., 2013; Laurent, private communication), and determine an improvement in the correction for the CO peak interference (zero error or spectral-baseline correction) due to the nearby H2O peak, with respect to the factory values. To this end, we use a simple and accurate novel method. We consider the equation:

$$p84sw' + A \cdot bh + B \cdot bh^2 + C \cdot bh^3 = p84sd' \cdot (1 + d \cdot bh + e \cdot bh^2) \qquad (14),$$

which is equivalent to Eqs. (12) and (13) except for the cubic polynomial on the left-hand side that accounts for the mentioned CO spectral-baseline correction. We advance that the novelty is not in the cubic correction (Chen et al., 2013; and Karion et al., 2013, used a quartic correction), but in the method described below. We use $d$ = -0.01287 and $e$ = -0.0005365 (Rella, 2010), and $p84sd'$, $A$, $B$, and $C$ are determined by least-square fitting of the experimental data to the cubic equation:

$$p84sw' = p84sd' + (d \cdot p84sd' - A) \cdot bh + (e \cdot p84sd' - B) \cdot bh^2 - C \cdot bh^3 \qquad (15),$$

obtained rearranging Eq. (14). The instantaneous CRDS CO signal is quite noisy, but the noise is significantly reduced by using 4000-data running means (39.2-minute approximately) without compromising the accuracy of the data due to the long duration of the experiment. Note that least-square fits are very sensitive to outliers, and the fit will be more accurate if the 4000-data running mean is performed previously (using a 39-minute running mean instead of a 1-minute running mean, the random noise is reduced by a factor of 6, approximately). As far as the authors know, this is new in the GHG monitoring literature. It is important to have a very accurate H2O correction, since in spite of the fact that the instantaneous CRDS CO values are quite noisy, such noise decreases significantly when performing successively 30-second, 1-hour, 12-hour means, whereas the hypothetical error in the H2O correction remains constant, behaving as a bias. After performing the 4000-data running means, we aggregate data computing 100-data means as for CO2 and CH4. Figure 11 shows $p84sw'$ versus $bh$ during the experiment and the least-square fitted curve corresponding to Eq. (15). After determining the coefficients of that cubic polynomial, we solve for the unknown constants in Eq. (14): $A$ = -5.287565 raw/pph$^{-1}$, $B$ = 5.283987 raw/pph$^{-2}$, and $C$

= -1.120169 raw/pph$^{-3}$. The absolute value of A is much smaller than 1000*|w1|, which appears in Eq. (5). This means $A$ is indeed a relatively small correction. On the contrary, $B$ and $C$ are very significant corrections comparing with their respective terms in Eq. (5), in which there is not cubic term. Figure 11 also shows the difference between the raw CO provided by the CRDS and the CO processed in the way described in this article (including H2O correction and calibration), both after performing a 4000-data running mean. The difference of -3.5 ppb at the intercept is due to the calibration (COraw is not calibrated), whereas the "sinusoidal" behaviour in $bh$ is due to the different spectral-baseline correction, which in the first variable is generic whereas in the second variable is specific for our CRDS unit.

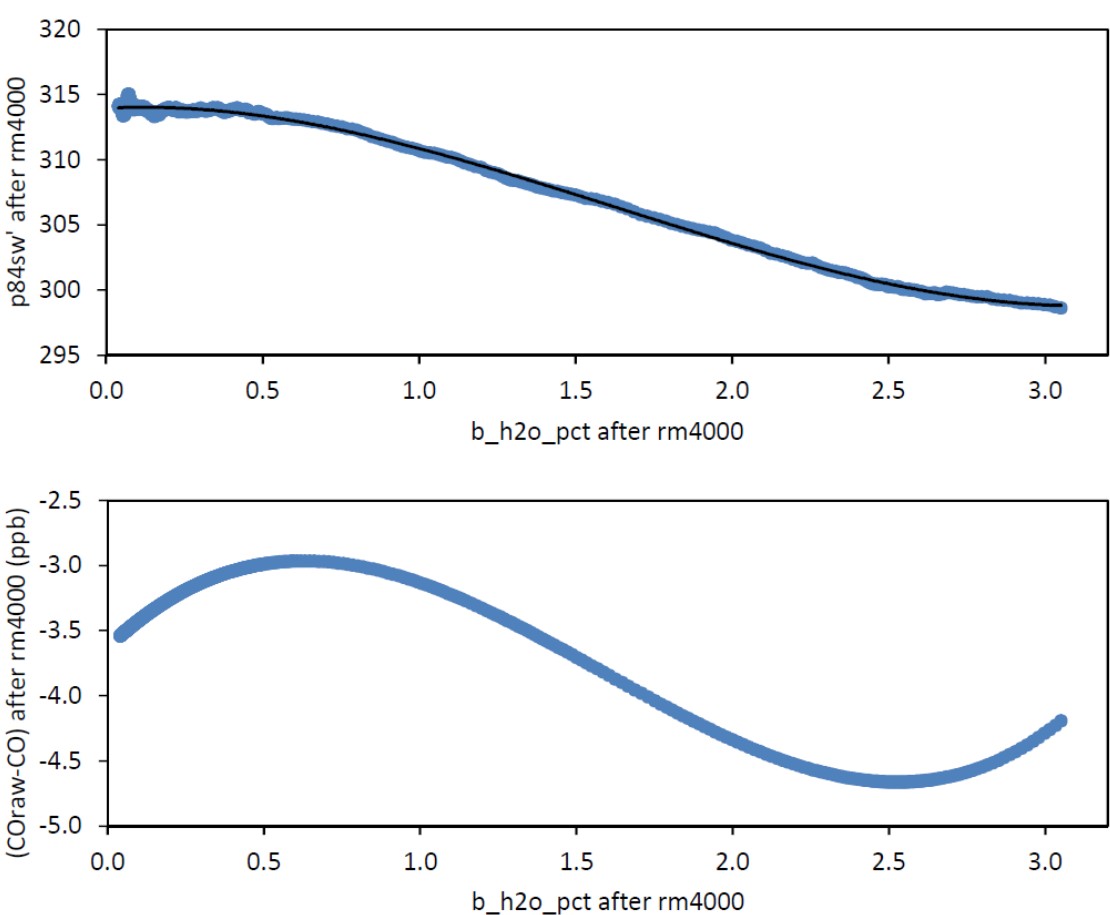

**Figure 11: Upper plot:** *p84sw'* **versus** *bh* **during the water droplet experiment (blue dots), both after performing a 4000-data running mean, and the least-square fitted curve corresponding to Eq. (15) in black. Lower plot: difference between the raw CO provided by the CRDS and the CO processed in the way described in this article (including H2O correction and calibration), both after performing a 4000-data running mean.**

## 6 Ambient Measurements

Figure 12 shows the ambient air/gas standard plumbing configuration operational since 28 November 2016. Before that date, there was no "Dedicated inlet", no drying (no cooled flasks), no solenoid nor needle valves, and ambient air entered through the MPV. Operational ambient air measurements started on 27 November 2015. Target gas measurements started on 18 December 2015 with a 7-hour cycle (3 hours of ambient, 30 minutes of target 1, 3 hours of ambient, and 30 minutes of target 2) to monitor better the behaviour of the CRDS, which became a 21-hour cycle after 24 June 2016. With the new plumbing configuration, ambient air is alternatively sampled from the two inlet lines within the 21-hour cycle (15 hours of ambient from the dedicated inlet and 5 hours of ambient from the general inlet). This has two purposes: 1) to provide plenty of time to exchange the flask used to trap $H_2O$ in the air line not used at this moment; 2) to check the consistency between both lines after every switch: a bias between them might indicate the existence of a leak in one of the lines (e.g., in the general inlet, which has a few large unions and several instruments connected to it; or at the flasks connections). Since the cooler bath temperature is -40 ℃, there is no complete drying. Therefore, it has been necessary to apply the water vapour correction for the full measurement period.

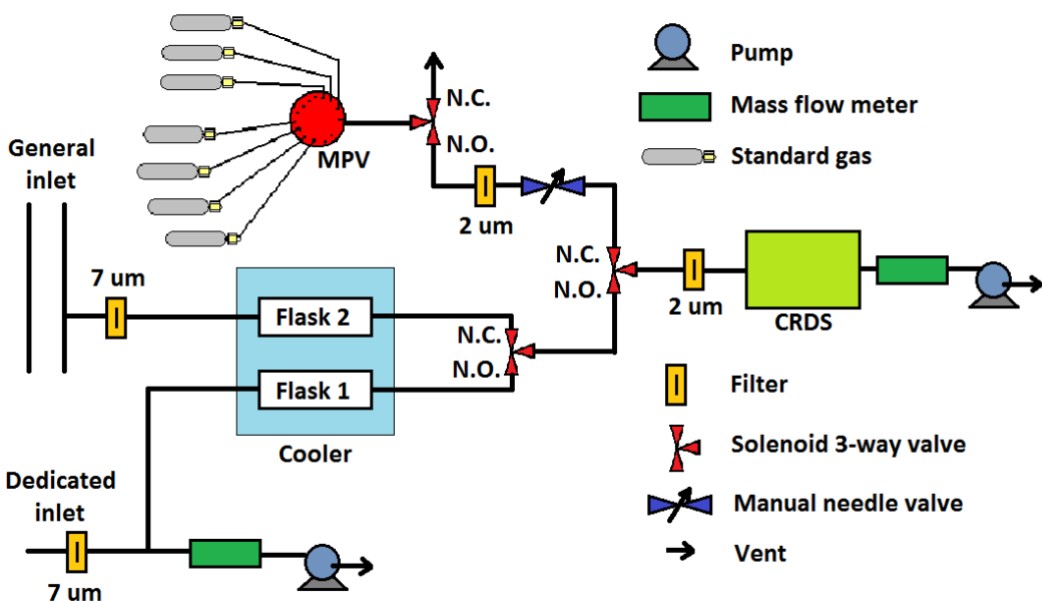

**Figure 12: Ambient air/gas standard plumbing configuration operational since 28 Nov 2016.**

### 6.1 Ambient Air Measurement Processing

After performing the pre-processing detailed in Sect. 3, we apply an additional filtering to the 30-second means. We retain a pre-processed 30-second mean if the following conditions are met: 1) the mean values of the following variables are within

the indicated ranges: *CP*: 186.7 ± 0.047 hPa (140 ± 0.035 Torr), *CT*: 45 ± 0.02 ℃, and OV: 20000-40000; 2) there is a calibration before and after the ambient mean considered, distant in time less than 180 days from each other, as is done in ICOS (Hazan et al., 2016).

Then, we apply the following processing scheme. Firstly, we apply water vapour correction: a) using Eqs. (12) and (13) we
compute CO2ovc_dry and CH4raw_dry from CO2ovc_wet, CH4raw_wet and *hr* (i.e., dilution and pressure broadening effect correction); b) using Eq. (14) we compute p84sd' from p84sw' and *bh* (i.e., refinement of the interference correction as well as dilution and pressure broadening effect correction). Secondly, we apply the calibration curves interpolated linearly in time: a) for CO2, we employ Eq. (9) using CO2ovc_dry where it says CO2ovc; b) for CH4, we employ Eq. (6) using CH4raw_dry where it says CH4raw; and c) for CO, we employ Eq. (7) using p84sd' where it says p84sw'. Finally, we
proceed to discard data due to the stabilization time after any sample path switch: 10 minutes for ambient measurements, and 20 minutes for target and calibration gas injections.

Figure 13 shows implicitly the H2O corrections applied to CO2 (dCO2_H2O), CH4 (dCH4_H2O) and CO (dCO_H2O) in physical units for ambient air measurements at IZO, e.g., dCO2_H2O is equal to (CO2ovc_dry – CO2ovc_wet) divided by the local slope of the calibration curve (Eq. 8). The plots for CO2 and CH4 are curves. However, the plot for CO shows a
cloud of dots because Eq. (14) is more complex and there does not exist a bijection between the X and Y variables.

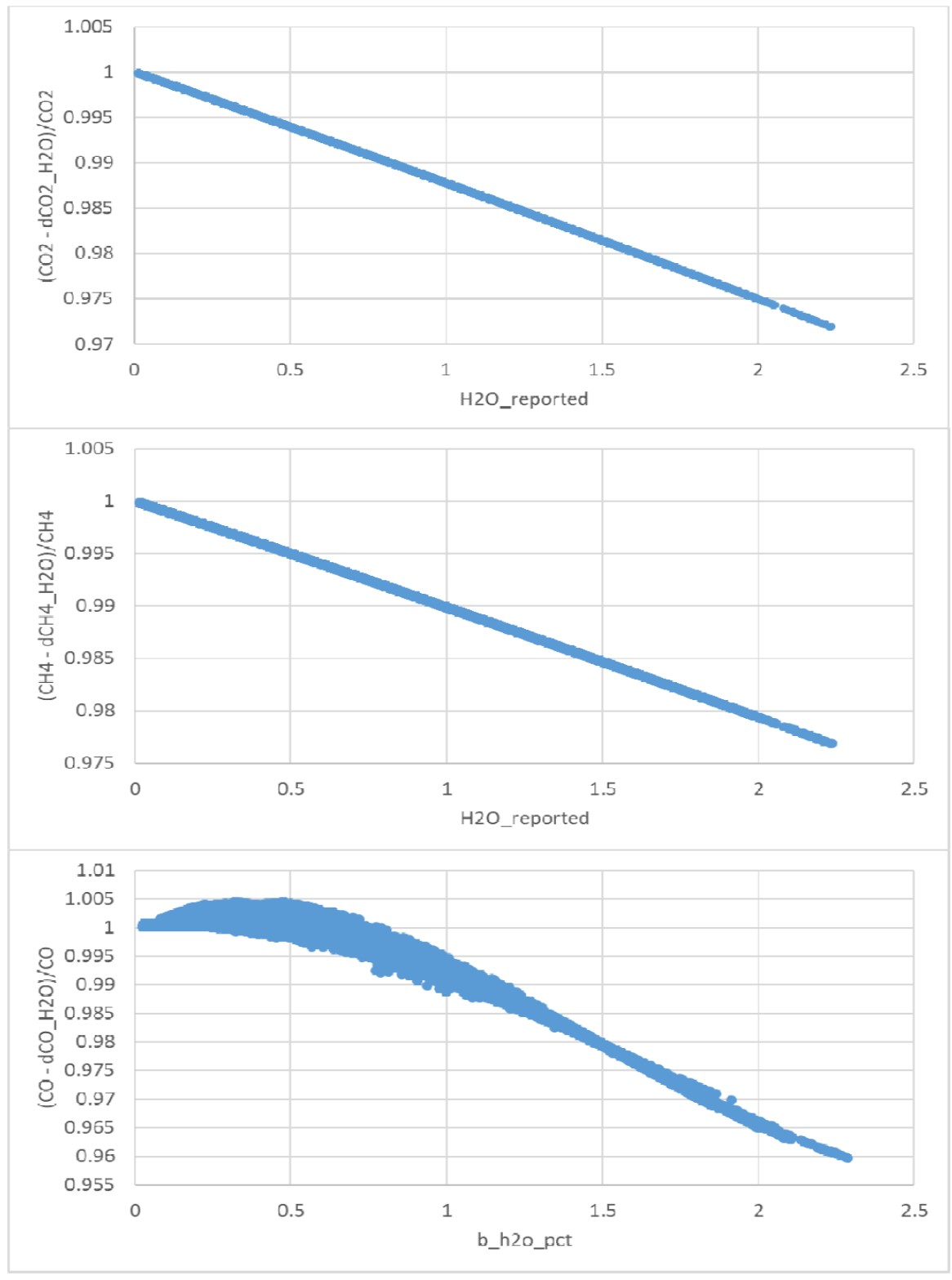

**Figure 13. Indirect plots of the H2O corrections applied to CO2 (dCO2_H2O), CH4 (dCH4_H2O) and CO (dCO_H2O) in physical units for ambient air measurements at IZO, e.g., dCO2_H2O is equal to (CO2ovc_dry – CO2ovc_wet) divided by the local slope of the calibration curve (Eq. 8).**

After the date in which the cold bath was implemented (-40 °C), the applied H2O corrections for ambient air measurements have been quite small: 1) for CO2, the mean correction has been 0.10 ppm (maximum: 0.36 ppm, minimum: 0.09 ppm; both for 10-minute means); 2) for CH4, the mean correction has been 0.38 ppb (maximum: 1.39 ppb, minimum: 0.34 ppb; both for 10-minute means); and 3) for CO, the mean correction has been -0.03 ppb (maximum in absolute value: -0.07 ppb, minimum in absolute value: -0.01 ppb; both for 10-minute means). Since those corrections are quite small, for this period

relying on the generic factory H2O corrections would have been sufficient.

### 6.2 Target Gas Injections

Table 6 summarizes the statistics for the target gas injections (30-second-mean assignments), whereas Fig. 14 shows the time series of mole fraction assignments for one of the target gases. The results are good. Note that 30-second is a too short time for CO, and it is necessary to consider longer averages for reducing noise. The ambient processing scheme described in Sect.

6.1 is also used to assign mole fractions to the target and calibration gas injections, and it is checked that the water vapour correction for them is smaller than 0.01 ppm for CO2, and 0.1 ppb for CH4 and CO.

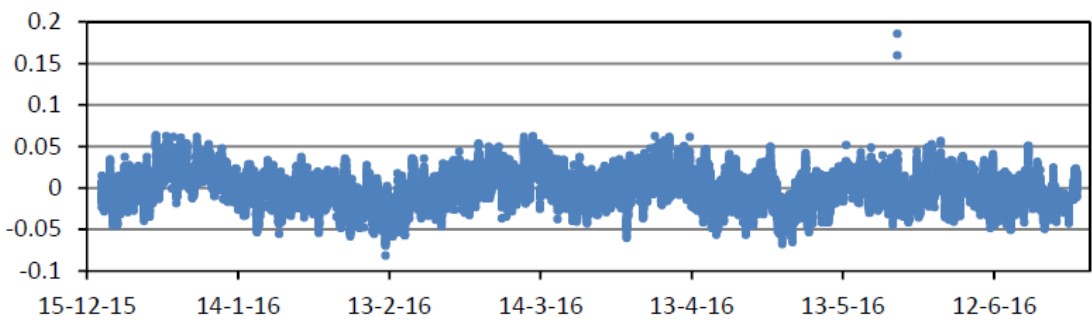

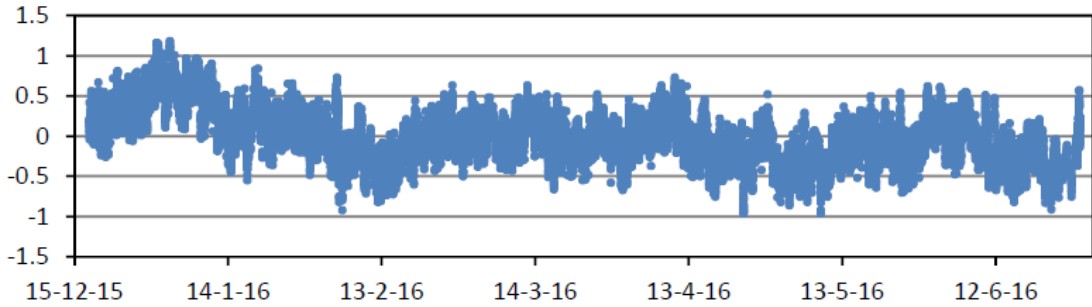

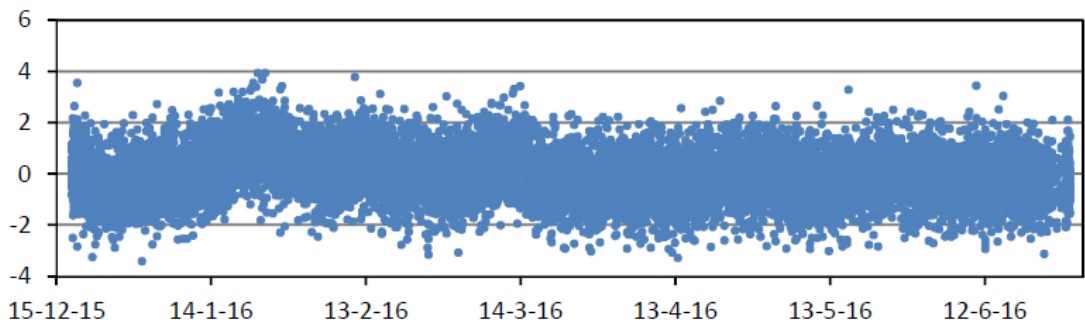

**Figure 14: Time series of mole fraction assignments for one of the target gases.**

| Tank/months | CO2 (ppm) | Std.dev. | CH4 (ppb) | Std.dev. | CO (ppb) | Std.dev. |
|---|---|---|---|---|---|---|
| CA07080/ 7m | 381.96 | 0.020 | 1825.43 | 0.32 | 148.60 | 0.97 |
| CA05038/ 7m | 368.85 | 0.020 | 1777.04 | 0.33 | 93.56 | 0.99 |
| CA06812/ 13m | 372.48 | 0.020 | 1784.80 | 0.27 | 142.04 | 1.01 |
| CA05034/ 13m | 363.71 | 0.020 | 1775.89 | 0.27 | 139.11 | 0.98 |

**Table 6. Statistics for the target gas injections (30-second-mean assignments): mean and standard deviation (Std.dev.) for the different species.**

**6.3 Comparison with other continuous measurements carried out at Izaña**

In this subsection, we compare the CRDS IZO ambient daily-nighttime (from 20:00 UTC of the previous day till 8:00 of the considered day) means with the co-located hourly means from the IZO Li7000 NDIR for CO2 (Gomez-Pelaez et al., 2011; Gomez-Pelaez et al., 2014; Gomez-Pelaez et al., 2016), IZO Varian GC-FID for CH4 (Gomez-Pelaez et al., 2011; Gomez-Pelaez et al., 2012; Gomez-Pelaez et al., 2016), and IZO RGA-3 GC-RGD for CO (Gomez-Pelaez et al., 2013; Gomez-Pelaez et al., 2016), all of them in the scales already indicated in Sect. 4. We use daily-nighttime means for the comparison because: 1) as mentioned in the introduction, IZO has background conditions during nighttime; and 2) when using 12-hour averages, we improve the signal to noise ratio and remove any hypothetical dependence on the used IZO general inlet due to small inhomogeneities in space and time of the mole fraction fields. Note that the data for 2017 are still not final. Figure 15 shows the time series of daily-nighttime CRDS measurements, whereas Table 7 shows the monthly-mean differences between the daily-nighttime CRDS measurements and those for the rest of the mentioned IZO instruments. As Table 7 shows, for CO2 and CH4 the differences between the instruments are within the GAW compatibility objectives (0.1 ppm for CO2 and 2 ppb for CH4), except for 4 months for the former and 3 months for the latter (coincident with those for CO2). The larger differences took place in October and November 2016. We think they were produced by a small leak in the general inlet used by the CRDS (the NDIR and the GC-FID used another general inlet). However, for CO the difference between the instruments is larger than 2 ppb after March 2016. The WMO tertiary standards used in the RGA-3 have been calibrated two times by the WMO CCL and the inferred drifts were considered significant, extrapolated forward in time, and taken into account in the RGA-3 data processing. These results seem to support the hypothesis that the observed negative differences are explained by the fact that the CRDS laboratory standards (WMO tertiaries) might be drifting up significantly for CO (see Sect. 4).

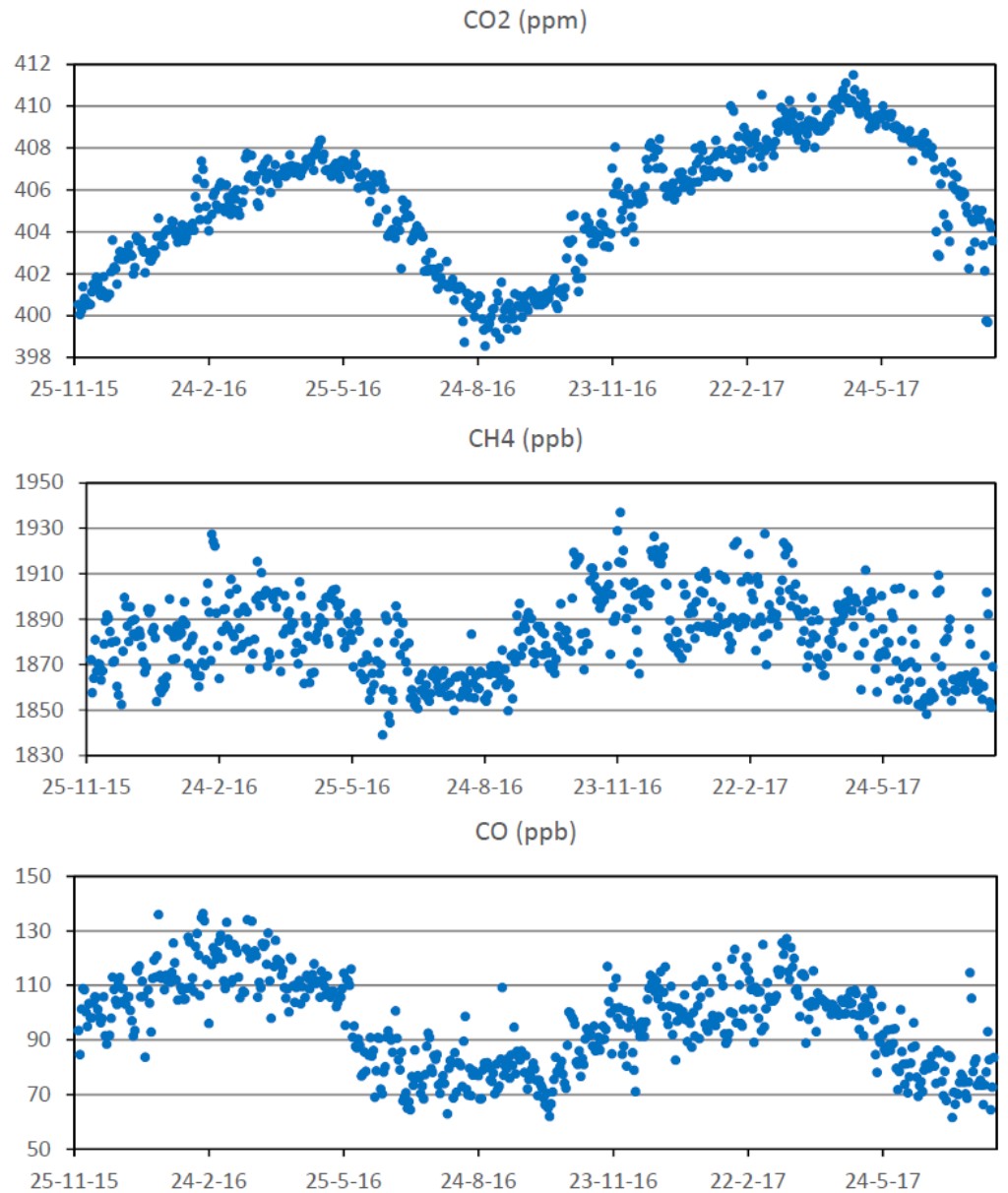

**Figure 15: Time series of daily-nighttime (12-hour averages) CRDS measurements (CO2, CH4 and CO).**

| Year | Month | CO2 CRDS - CO2 Li7000 (ppm) | CH4 CRDS - CH4 VarianFID (ppb) | CO CRDS - CO RGA3 (ppb) |
|------|-------|------|------|------|
| **Full** | **period** | **-0.07** | **1.2** | **-2.8** |
| 2015 | 11 | -0.08 | 1.6 | -0.4 |
| 2015 | 12 | -0.04 | 0.9 | -1.1 |

| 2016 | 1 | -0.03 | 1.0 | -1.3 |
|------|----|-------|------|------|
| 2016 | 2 | -0.08 | 1.0 | -1.8 |
| 2016 | 3 | -0.07 | 0.3 | -2.4 |
| 2016 | 4 | -0.06 | -0.1 | -2.7 |
| 2016 | 5 | -0.02 | 0.2 | -2.9 |
| 2016 | 6 | -0.03 | 0.9 | -3.0 |
| 2016 | 7 | 0.00 | 1.1 | -3.2 |
| 2016 | 8 | 0.05 | 1.9 | -2.9 |
| 2016 | 9 | -0.10 | 2.0 | -3.1 |
| 2016 | 10 | -0.12 | 3.3 | -4.1 |
| 2016 | 11 | -0.15 | 3.1 | -3.0 |
| 2016 | 12 | -0.07 | 0.5 | -3.7 |
| 2017 | 1 | -0.09 | 1.1 | -3.0 |
| 2017 | 2 | -0.14 | 2.1 | -3.3 |
| 2017 | 3 | -0.11 | 0.5 | -3.6 |
| 2017 | 4 | -0.08 | 0.4 | -2.7 |
| 2017 | 5 | --- | 0.3 | -3.4 |

**Table 7: Monthly-mean differences between the daily-nighttime CRDS measurements and those for the rest of the mentioned IZO instruments (Li7000 NDIR for CO2, Varian GC-FID for CH4, and RGA-3 GC-RGD for CO).**

.

**7 Preliminary independent assessment on the drift rates of the CRDS CO standards**

The evidence presented in Sect. 4 and Sect 6.3 seem to indicate the CRDS CO WMO tertiary standards might be drifting significantly. These standards have been calibrated only once by CCL: in August-September 2015. In order to perform a preliminary independent assessment on the drift rates of the CRDS CO standards, we have proceeded as follows. As mentioned in Sect 6.3, the WMO tertiary standards used in the RGA-3 have been calibrated twice (9 years distant) by the WMO CO CCL and the inferred drifts were considered significant and extrapolated forward in time. We have performed a 4-

cycle calibration in the CRDS to compare the CRDS standards (CB11240, CB11389, CB11393, and CB11340) and the RGA-3 standards (CA06968, CA06768, CA06988, CA06946, and CA06978). The first step has been assigning CO2 and CH4 mole fractions to the RGA-3 standards using the calibration curves obtained using the CRDS standards. The second step has been assigning CO mole fractions to the CRDS standards using the calibration curve obtained using the RGA-3 standards, whose fit RMS residual is 0.4 ppb. In the third step, we determined the drift rate of each CRDS standards using

the CCL assignment done in 2015 and the present assignment (done on 4 October 2017), which is indirectly traceable to the

WMO primaries. The standard CB11240 is the only one with a significant drift rate, 1.21 ppb/year, having 195.87 ppb at present. The other three standards all have positive drift rates, with the maximum being 0.17 ppb/year. We have performed the exercise of reprocessing all the CRDS calibrations taking into account the drift rates determined for the four CRDS standards (even those which are not statistically significant), as shown in Fig. 16. Comparing Fig. 16 with Fig. 7c, we see now there is no trend in the RMS residual from the calibration fit and the downward drift of the target gases is significantly smaller. However, when reprocessing the CRDS ambient CO time series, the maximum improvement in the CRDS minus RGA-3 monthly difference time series is 0.3 ppb for some periods, remaining unchanged the global mean difference for the full period. Therefore, we infer the performance of the calibrations improves largely when taking into account the quite significant drift in the standard CB11240, but the CRDS versus RGA-3 ambient differences remain almost unchanged. Perhaps, a possible explanation might be in the problems detected recently by the CCL in the CO WMO-X2014A scale (https://www.esrl.noaa.gov/gmd/ccl/co_scale_update.html).

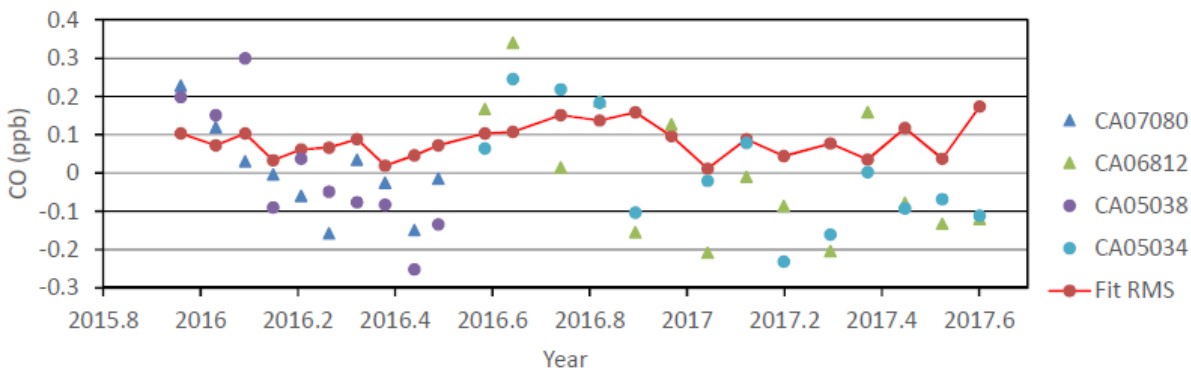

**Figure 16: RMS residual of the fit (red dots/line), and difference between the assigned mole fraction to a target gas and the mean mole fraction of such target gas (a different colour is used for each target gas), for each species and mole-fraction calibration performed using 4 WMO laboratory standards (every 3 or 4 weeks), after taking into account the drift rates of the CRDS standards.**

**8 Summary and Conclusions**

- At the end of 2015, a CO2/CH4/CO CRDS was installed at the Izaña Global GAW station to improve the Izaña GHG GAW measurement programme and guarantee its long-term maintenance. The CRDS passed the acceptance tests. However, a correction for CO2 that takes into account the inlet pressure had to be incorporated in order to achieve a RMS residual of around 0.02 ppm, which is the value we obtain with the IZO NDIR based measurements systems. For CO, our data processing is based on the raw spectral peak data instead of on the raw CO provided by

the instrument. The relationships between flow rate, CRDS inlet pressure and CRDS outlet valve aperture have been determined also. The CRDS inlet pressure sensitivity is determined for the different compounds as well as its stability over the years.

- We use linear response functions for CH4 and CO, and a quadratic response function for CO2. The CRDS long-term drift of the raw responses is: 0.104 ppm/year for CO2, 2.22 ppb/year for CH4, and 0.544 ppb/yeur for CO. Assuming that those drifts are due to drifts in the real pressure and temperature of the cavity, we have determined that our CRDS has a long-term drift of 0.0678 ºC/year and 0.595 hPa/year in the cavity sensors using relations between partial derivatives. We show also the evolution in time of the response-function local slopes at the mole fractions of the virtual tank, as well as the RMS residual in the calibration fits, which has no significant trend except for CO.

- The time series of target gas assignments during calibrations are also shown, which again indicate a good behaviour for CO2 and CH4, but a downward drift for CO. Those facts suggest the CRDS CO WMO tertiary standards are probably drifting significantly in spite of the fact they have been only used during two years. Using an independent set of CO laboratory standards whose drift rates have been determined by the CO CCL, we conclude that one of the CRDS standards is drifting quite significantly (1.21 ppb/year). The performance in the calibrations improves when taking that drift into account.

- The results of the long water-droplet test (12 hours) have been presented and used for the H2O water vapor correction. The determination of the H2O correction for CO presents two novelties: use of the raw spectral peak data and use of a running mean to smooth random noise before performing the least square fit.

- We have presented the ambient measurement scheme and its data processing. Target gas injections show very small standard deviations except for CO. The agreement with other IZO in situ continuous measurements is good most of the time for CO2 and CH4, but for CO is just outside the GAW 2-ppb objective. It seems the disagreement is not produced by the drifts in the CRDS CO WMO tertiary standards. The mean differences for the full period are: -0.07 ppm for CO2, 1.2 ppb for CH4, and -2.8 ppb for CO.

- We have determined and discussed the physical origin of the inlet pressure and H2O dependences of the CRDS flow, and pointed out the existence of flow-rate-dependent small spatial inhomogeneities in the pressure and temperature fields inside the cavity. We have shown that the slightly-depleted-in-pressure regions inside the CRDS cavity in the neighbourhood of the inlet and outlet pipes due to the cross-section change, are probably the cause of the slight CO2 correction associated with the mass flow rate we have empirically obtained. We suggest performing a gas dynamic numerical simulation of the pressure and temperature fields inside the CRDS cavity for different flow rates. This could help to improve the spectral forward model used by the CRDS and also to take into account more accurately the impact of the flow rate on the measurements. Furthermore, the use of conical adapters for connecting the pipes to the CRDS cavity might keep the pressure gradients associated with the cross-section changes out of the laser path.

**Code availability.** The Fortran 90 codes developed in this work could be made available to other researchers under a cooperative agreement with the Izaña Atmospheric Research Centre (AEMET). However, a very limited support on their use could be provided.

**Data availability.** The data presented in this paper is available under request. If the supplied data is intended to be used in a scientific article, co-authorship should be offered to data providers.

**Appendix A. A note concerning the H2O dependence of the CRDS flow and the spatial inhomogeneity of the pressure field inside the cavity**

The theoretical equation relating the standard volumetric flow ($F$) with the inlet quantities for a choked flow is presented and

discussed in Sect. 2.4. However, additionally to the dependences on pressure and temperature, there is also a small dependence on the water vapour mole fraction through the ratio of specific heats and the gas constant, which depend slightly on the H2O mole fraction as follows:

$$\frac{1}{R} = \frac{1-r}{R_d} + \frac{r}{R_{H2O}} \qquad (A1),$$

where $r$ is the H2O mole fraction in mol/mol, $R_d$ is the gas constant for dry air, and $R_{H2O}$ is the gas constant for H2O;

$$\gamma = \frac{1.4+0.4 \cdot r}{1+0.4 \cdot r} \qquad (A2).$$

In order to estimate the relative impact of water vapour changes on the standard volumetric flow, we need to know approximately the values of $C_d$ and $A$. To this end, we use this fact obtained from Table 3: when $p_i$ is 700 hPa, $F$ is 150 scc/min, and obtain using Eq. (2) that the product of $C_d$ and $A$ is equal to 1.955e-8 m². Figure A1 shows $F$ and its derivative (with respect to $r$) as functions of $r$, for $p_i$ = 700 hPa and $T_i$ = 293 K, showing that the relative impact of $r$ on $F$ is small and

the derivative is quite constant for the considered $r$ range (0.00-0.05).

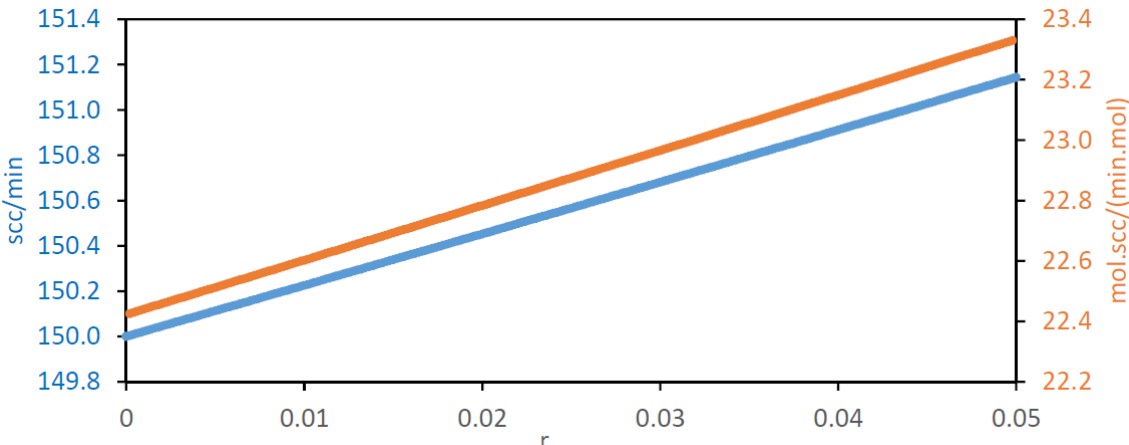

**Figure A1: Standard volumetric flow (*F*) on the left Y-axis and its derivative with respect to r (H2O mole fraction in mol/mol) on the right Y-axis, for $p_i$ = 700 hPa and $T_i$ = 293 K.**

When there is a stationary air flow through an instrument, the pressure field changes spatially mainly due to two reasons (Bernoulli equation): 1) longitudinal decrease of the pressure due to viscosity, and 2) changes in the cross-section along the pipe, which require flow acceleration (provided by a longitudinal pressure gradient) when the cross-section decreases and flow deceleration when the cross-section increases (e.g., Venturi effect). The structure of the optical cavity of the CRDS G2401 is shown in Fig. 1 of Crosson (2008). The plane defined by the three mirrors inside the cavity is horizontal (parallel to the surface of the Earth when the instrument is set on its feet on a bench). The inlet and outlet cavity ports are on the top of the cavity. The pressure sensor is on a third port located on the top of the cavity, at the approximate centre (Rella, private communication). Applying considerations of fluid dynamics, we infer the following facts. First, along the sense of flow inside the cavity, there needs to be a very small decrease in pressure in order to be able to balance the resistance to flow due to viscosity, and that decrease will be larger as the mass flow rate is increased. Since the pressure sensor is located in the middle of the cavity, the mean pressure will be monitored. A parcel of fluid flowing along the cavity will expand very slightly (due to the decrease in pressure that the Lagrangian parcel experiences), and therefore, the temperature will tend to slightly decrease adiabatically in all the points of the volume whereas the heat to compensate it comes from the surface of the cavity. That is, necessarily there needs to be also a small inhomogeneity in the temperature field inside the cavity, and this inhomogeneity depends on the flow rate. The hypothetical net effect on the measurements is difficult to assess a priori without performing a gas dynamics numerical simulation. Second, when a fluid parcel leaves the inlet pipe and enters into the cavity, it experiences a large change in the cross-section of the solid material that contents the flow. Therefore, there needs to be a portion of cavity near the inlet with pressure increasing in the flow sense to decrease and accommodate the velocity of the fluid. That is, in the cavity near the inlet, there is a pressure smaller than in the rest of the cavity. Moreover, the opposite process happens in the cavity near the outlet: the fluid needs to be accelerated, and therefore, there needs to be a portion of cavity near the outlet with pressure decreasing in the flow sense. That is, in the cavity near the outlet, there is a smaller pressure than in the rest of the cavity, as happens near the inlet, and this decrease is larger when the mass flow rate is increased. If any portion of those two regions is crosses by the laser path, the perturbation this produces in the measurements agrees in sign with Eq. (3). Therefore, this might be the explanation of the empirically observed effect described in Sect. 3.1. All the effects pointed out in this appendix are new in the GHG measurement literature according to the knowledge of the authors.

**Appendix B. CO2ovc_raw versus *hr* during the water droplet experiment**

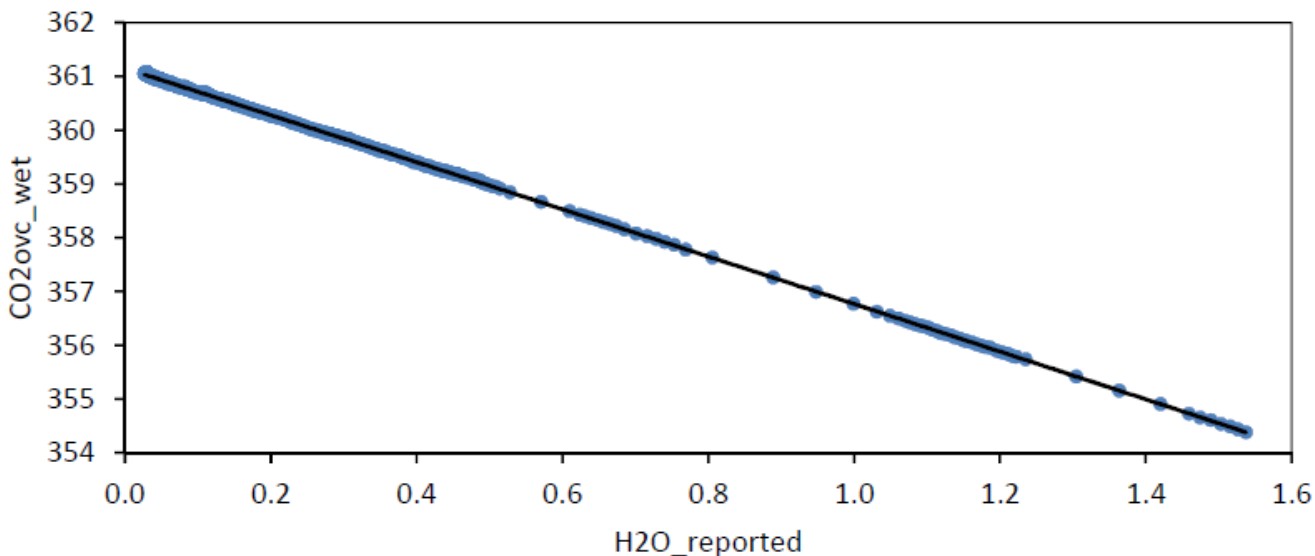

**Figure B1: CO2ovc_raw versus *hr* during the water droplet experiment (in blue) and the least-square fitted curve (in black), where only the 0.0-1.55 *hr* range has been plotted.**

**Appendix C. Some additional novelties in the Izaña GHG instrumentation since GGMT-2015**

We have introduced the following improvements in the dedicated inlet lines of the IZO GHG measurement systems since GGMT-2015: 1) backpressure regulators for the vents located downstream of the pumps, and rotameters for those vents; 2) needle valves in low flow vents installed downstream of the cryotraps; 3) glass flask cryotraps with Ultra-Torr connections; and 4) hermetic plugs for unused ports of the rotary Valco valves.

We have prepared two CO2 laboratory standards of 418.7 ppm for the Izaña NDIRs Li7000 and Li6252 (using two cylinders that have proved to be very stable in previous uses as CO2 working standards), and calibrated them against our CRDS WMO laboratory standards using the G2401 CRDS.

We have reprocessed the Izaña time series of CH4 and CO in the scales X2004A and X2014A, respectively, taking into account also the drift of the five WMO laboratory standards used in the Izaña RGA-3.

**Author contribution**. A.J. Gomez-Pelaez designed the measurement system, measurement scheme, and response functions, made the CRDS acceptance tests, configured the CRDS, determined the relationships between flow rate, OV and CRDS inlet pressure, as well as its impact in the CRDS measurements, performed the H2O droplet tests, wrote the numerical codes, analysed the data, made the study of Appendix A, wrote the manuscript and made the plots. R. Ramos installed the measurement system, helped in the routine operation of the system, revised the manuscript, and performed the additional

inlet pressure sensitivity tests carried out in 2018. E. Cuevas was the PI of the financed R+D infrastructure project by which the CRDS equipment could be purchased and installed at IZO, and revised in detail the manuscript. V. Gomez-Trueba performed the routine calibrations, and helped perform the additional inlet pressure sensitivity tests carried out in 2018. E. Reyes provided support configuring the CRDS computer, prepared the external media necessary to perform daily copies of

the acquired data, and revised the manuscript.

**Competing interests**. The authors declare that they have no conflict of interest.

**Acknowledgements**. The acquisition of the instrument was largely financed by European ERDF funds through the Spanish

R+D infrastructure project AEDM15-BE-3319 of the Spanish "Ministerio de Economía, Industria y Competitividad". This study was developed within the Global Atmospheric Watch (GAW) Programme at the Izaña Atmospheric Research Centre, financed by AEMET. We thank Chris Rella (Picarro, Inc., USA), Christoph Zellweger (EMPA, Switzerland), and Olivier Laurent (ICOS ATC/ LSCE, France) for providing information of great interest about the Picarro G2401 CRDS, and to the Izaña observatory staff. We acknowledge the constructive comments of the anonymous referees.

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
