# Peer review of "Atmospheric CO2, CH4, and CO with CRDS technique at the Izaña Global GAW station: instrumental tests, developments and first measurement results"

_Atmospheric Measurement Techniques, 2017_

## Referee Comment (RC1) · Anonymous Referee #1 · 30 Nov 2017

This work describes the GHG measurement system at Izana Observatory (IZO). Several novel measurement methods or improvements to standard methods are described, including a new correction to CO2 in Picarro analyzers that is caused by inhomogeneous mixing in the measurement cell of the analyzer that is affected by flow rate. It should be published after corrections outlined below.

One major comment that should be addressed is the determination of the co2 correction for inlet pressure variation should be tested more. Details and recommendations are below. A second is that the authors use a few more complex fits and equations for

[Figure]

corrections than reported previously in the literature for these same instruments, and there is little justification or evaluation of the improvement that these complex corrections make on the data, or reduction in error/uncertainty (e.g. quadratic fit to the CO2, and water vapour corrections).

Major & Minor comments:

Acronyms: Some are introduced without definition in the abstract (CO2, WMO); Others esp. in the introduction are defined backwards from the usual convention, i.e. the acronym comes first with the expansion in parentheses. But not consistently. And some are introduced (WMO) prior to definition. The authors should carefully go through the text and fix these.

L29: awkward (requirements are higher as higher the lifetime...). Requirements are more stringent for gases with a longer lifetime perhaps?

P2, L4: American billion?

P2, L4, ppm should be defined here as micromoles per mole of dry air, or dry mole fraction. (and similar for ppb).

P2 L8 awkward: "therefore, the required frequency... is much lower".

P2 L30 Tertiary shoudl not be capitalized I don't think.

P2 L32 until now

P3 L12 "being" should be ", rejecting the first hour as stabilization time."

Table 1 caption should say "for two averaging times". Some note should be made explaining the negative water vapour values here.

Table 2 - wouldn't this be a more appropriate place to report the average H2O? It would be good to know if the water vapour had completely left the stream by the time these SD's were taken. Some explanation of the higher SD's over these 10 minutes compared to the Table 1 results should be made - perhaps the cylinder is still "stabilizing", but then did the authors see a trend in the measurement over the 10 minutes? Was any such trend or noise caused by water vapour still drying out?

P4 L4: How was the ambient pressure test conducted? This paragraph should have more detail (pressure chamber?). I agree the effect of ambient pressure is small - you could note that this analyzer is not meant for use on aircraft where large pressure changes might occur. But I still wonder if this effect on the measurements is transient or dependent only on the absolute ambient pressure. Yver Kwok et al state that they find no ambient pressure dependence on $CO_2$ and $CH_4$.

P4 L10: these pressures hould be given in millibar or SI units - presumably this is gauge pressure, psig. Absolute pressure would be better, perhaps stating the regulator gauge pressure in parentheses. Was the effect only noted for these two pressures, or was there more testing done at intermediate pressures? Was it repeatable, or are these numbers given only for the single test between two pressures? (this is a question I had later, on how the fit was made to the outlet valve - was the fit based only on two points, and was it evaluated for a repeated test?).

p4 L23: "during that first calibration" - what calibration? perhaps "During the calibration described above,"

P4 L27, awkward phrasing "to the computation". Perhaps just "Pre-processing refers to the computation of raw data 3-second means... as well as the computation of some derived variables". (what variables?). Also "no" should read "not".

P5 L3-10. It is not clear why certain variables are associated with certain species codes. I thought that the species code had to do with which laser was making the measurement (or the frequency range it was covering). One sentence explaining this perhaps would be useful to the reader. It might not be useful to mention the CO2_dry and CH4_dry variables, as they confuse things and they are not used in this work at all - they get updated after the h2o_reported scan. I find the explanation of each variable

useful however. I think CO2_dry is erroneously mentioned twice: once with sepcies=2 and once when species=3.

Why are both the MPV and SV fields used - should mention why these are monitored - presumably your gas handling uses both solenoid and MPV valves controlled by the Picarro software?

P5 L19, I believe the flow in the orifice itself is sonic, and is supersonic immediately downstream, depending on the ratio of upstream to cavity pressure. Also pressure units should be in mb or other SI, with torr in parentheses.

P5 L20 was this linear relationship between OV and inlet pressure confirmed in your tests? This should be investigated and shown, not just assumed, as the inlet pressure/ flow correction is using the OV variable as a direct proxy for flow rate essentially.

P5 Eq 1 - it would be good to see some more information on this fit. Was it really based only on the two different pressures investigated in the test? Could a fit to more pressures be performed? how good or bad is the fit - what are the residuals from the test itself? This is one of the main findings of this work, and has not been reported before, as the authors point out, so more work is needed to show the reader how the correction was determined, how it performs, and its consistency over time and after a reboot. The fit should be tested by performing a test again and checking for consistency and for errors after the fit was applied, esp. its reliance on the OV variable.

I do believe that because the inlet orifice is not a physical orifice but rather a proportional valve fixed at a certain opening, after the analyzer has a power cycle (reboot), it may not come back to the same spot exactly (the valve is not so precise, so given the same voltage might open to a different diameter slightly), and following that, the OV may not be in the same spot, i.e. for the same inlet pressure as prior to the power cycle, the OV value may differ. I think that this would render equation 1 incorrect and would need to be re-calculated. This should be checked and determined. It may be necessary to install a pressure sensor upstream of the CRDS instrument and perform

a linear fit on that rather than relying on the OV. This may not be true if there is a physical critical orifice upstream whose aperture never changes. The aircraft analyzers do have a physical critical orifice so that they can maintain a constant given flow rate, even after the instrument is rebooted.

p6 L22: perhaps rephrase to : "provided results that satisfied our accuracy requirements".

P6 L24: This helps avoid...

P6 L25: that might propagate to the interior of the cylinders...

P7 L6: What reason is given for going to a quadratic fit for CO2 here? In other literature (including Yver Kwok et al) a linear fit is generally used. Perhaps the authors can justify the use of a more complex curve for this instrument. It is useful to note that the residuals of a calibration should be compared to the assigned 1-sigma uncertainty of the gas standard relative to the scale. For NOAA tanks, this is something close to 0.02-0.03 ppm (as cited in Andrews et al., but could be checked for specific cylinders perhaps?). You wouldn't expect the residuals to be better than that.

P7 L6, L9: I would suggest replacing "real dry mole fraction" by "the dry mole fraction assigned by the CCL on the WMO scale".

p7, L16 "high-accuracy"

p7, L17 should read: the calibration fits are generally performed using a limited range...

p7 Is the virtual tank concentration its assigned value (x-axis) or measured value (y-axis)? Presumably, following the convention of equations 4-6, it is the x-axis, assigned value that is constant at 400 ppm and the raw response for that value is what is calculated using the equations. Perhaps just state this in the text when you mention the tank values.

p8-9: explain that the drift you are referring to is drift in the P, T sensors that then

causes the cavity to be controlled at a slightly drifting temperature. P9, L1: Was this supposed to read .152 degrees C / year rather than Torr?

Fig 2: is the slope of the red line here equal to the left hand side of equation 8?

Eqn 8 & 9: A little more information on this would be good. Does this analysis assume that Temperature drift is a linear function of pressure drift? I guess if they are both linearly drifting with time, this makes sense.

What is the slope? (Yver Kwok report a mean of 2.4). Perhaps a plot of the fractional change rather tahn the plot in Figure 2 would be more applicable to explain equations 8 & 9 (similar to figure 14 in Y-K).

p9: The singular of species is species, so I think it should be "for each species" (here and throughout the paper).

Fig 3: the Y-axis here is odd in terms of units. Not sure what to do about it.

p11: L14: explain "since CO standards use to drift upward"? When?

p11 L15: why at different rates? (or perhaps different rates from the target gas). If the 4 tertiary standards are drifting at different rates, one would expect residuals to be increasing in time on the fits as well? It seems that without further evidence, it might be more appropriate to just state that is is likely that the standards and/or target tanks are drifting.

Fig4: the labels should be on the Yaxis rather than as titles. Not sure if this was stated earlier, but each of these points represents an average of how long and every how often? This could be mentioned in the caption to remind the reader.

Figure 5: I don't find this figure particularly useful. If kept, labels should be put in the photo of the various components.

Figure 6: y-axis is unlabeled, should have the variable even though there are not units on this raw measurement.
p14 L24: 39 minutes seems to be very long, when looking at figure 6, the water vapour could change significantly in that time, especially at the higher H2O values at the beginning of the test. Is such a long average really required, when you are achieving 1-minute precision of 0.87 ppb?

Also, given the complex equations presented here for CO water correction, how different is the final result from the internal correction (i.e. the variable "CO" in the raw output)? It would be necessary for the reader to evaluate this and how it depends on the level of water vapour, to see if it's worth implementing.

P15-16, Regarding the -40C trap, can you relate this to a maximum H2O value you see in the data after that point? What is the magnitude of the water correction at those values, and how different is it to CO2_dry and CH4_dry? Rella et al. indicate that at low water vapour values, the internal correction works well. Again, the water vapour tests are cumbersome, especially for field analyzers, in many cases - it would be useful to the readers to know what kind of improvement is gained from the tests as opposed to using the factory correction in cases where there already is partial drying like this.

P16, L7-8: Awkward phrasing "for not discarding". Perhaps "We retain a pre-processed 30-second mean if the following conditions are met: ". Then rephrase the conditions accordingly (i.e. "the mean values of the following variables are within...") etc.

L9: condition 2 awkward: was exit supposed to be "exist"?

P16 L17: clarify when 10 minutes are discarded (after a calibration or any sample path switch?).

P18 L14: remove the final "in space", as earlier this sentence stated in space and time.

Fig 11, caption should include the fact that these are 12-hour averages (if that is the case).

Table 4, first row, month says 2017? I wonder if the information in this table would be better expressed in a figure somehow, with dashed lines indicating the WMO compatibility goals?

P21 L4: shown

In section 6.3, the reason for the CRDS-RGA differences for CO is ascribed as due to the CRDS standard drift, but here in Section 7, it is indicated that the drift did not improve the comparison. Can the authors give any explanation for this? Even if the RGA tanks are also drifting, the correction described in Section 7 would put both data sets relative to the same tanks after all.

Finally, overall, given the large effort involved in achieving very high accuracy for this data set, do the authors envision prescribing an uncertainty to their measurements? [perhaps this is outside the scope of this paper, but something to think about maybe for the future].

---

## Referee Comment (RC2) · Anonymous Referee #2 · 1 Dec 2017

In this paper, the authors present their instrumental set-up to measure greenhouse gases continuously at Izana global GAW station. The instrument performances are tested and interferences are studied. First ambient air measurements are compared against the historical NDIR and GC systems. This paper should be published after corrections as detailed below.

General comments:

The paper would benefit from an English proofing.

[Figure]

Specific comments:

p4: It would be great to have a plot of the inlet pressure tests. Did you only varied between two pressures? Also, it would be good to make this test several time to see if drifts occurs over time and if the results are reproducible as you are using them to correct data.

p4: You mention using smaller inlet pressures differences, does that lead to smaller RMS residuals? Is then the inlet pressure correction useful?

p5 l25: From experience, the outletvalve value change with time, depending on the Tdas temperature, after restarting the instrument or when the filters get clogged. It seems safer to operate as you said yourself by reducing the difference in inlet pressure between cylinders and ambient air to avoid the need for an empirical correction.

Section 5: did you perform more than one test to assess the variability? It would be interesting to plot the biases between the assigned dry value and the wet values depending on $H_2O$ for Picarro and your own correction for the three species. On the same subject: p15 l22: what is the level of residual water? Why not invest in a -60°C cryocooler and get rid of any correction as you are using already a crycooler system? Especially if you refer yourself to the paper of Reum et al. in discussion in AMT (Reum, F., Gerbig, C., Lavric, J. V., Rella, C. W., and Göckede, M.: An improved water correction function for Picarro greenhouse gas analyzers, Atmos. Meas. Tech. Discuss., https://doi.org/10.5194/amt-2017-174, in review, 2017.) that shows that the $H_2O$ correction is biased when almost dry due to the sensitivity of the pressure sensor with $H_2O$.

Section7: Can the ambient air difference be due to the non-linearity of the RGA-3 or to the fact that the $H_2O$ correction is not good enough? The difference does not seems to increase strongly over time but more to vary around a bias.

Technical comments:

p1 l29: not clear, please rephrase

p2 l7 replace "being.. much lower" by reducing

p2 l3 replace "ones" by "techniques"

p2 l26: add "of" after "physical discussion"

p2 l27: "as follow"

p3 l6 "serial number" should go before the actual serial number

p3 l7: remove "to the CRDS"

p3 l9: cite as well here Yver Kwok et al. As the tests are described in this paper while the specifications gives the tresholds.

p3 l11: In Yver Kwok et al. the terminology for the precision test is continuous measurement repeatability (CMR), rephrase for example as "The first test, defined in Yver Kwok et al; as the CMR test consists..."

p3 l12 put "being" after "the first hour". This type of exchange appears throughout the text, please check.

p3 l15: Replace precision by CMR

p3 l18: this test is what was called reproductibility before and in ICOS-ATC LTR or long term repeatability. Add ATC after ICOS as ecosystem or ocean could have defined other terms.

p3 l24: thresholds

p4 l2 Replace "Repeatability test" by LTR

p4 l5 use proper units, mbar not mb

p4 27-28 delete "to" before "the computation"

p6 l2 delete "tot" before "the raw CH4"

p6 l12-13 you multiply p84sw by 1000, then p84sw' should be in ppb contrary to what is mentioned.

p6 l28: numerical

p7 l28 It is a new paragraph and the transition is not smooth, please add a transition.

p11 l15 Refer to section 7 for more investigations.

p14 l 7-8: move "being" after "CH4", add "s" to coefficient

p15 l7 what is pph?

p15 l16: why is there two inlets, what is the purposes of switching between the two every day?

p16 l8: Rephrase "For not discarding" in "To keep"

p18 l18: Can you comment on the reason of the larger differences for CO2 and CH4 for these particular months? As it is both for CO2 and CH4, it seems to indicate that the CRDS would be the cause?

p20 l10: Replace "for comparing" by "to compare"

---

## Referee Comment (RC3) · Anonymous Referee #3 · 26 Dec 2017

Gomez-Pelaez et al. present both laboratory and field test results of a commercially available CO2/CH4/CO cavity ring-down spectrometer (CRDS). The authors discussed the results within the context of several relevant international programs, e.g. Global Atmosphere Watch (GAW) and the European Integrated Carbon Observation System (ICOS). Being aware of recent development of greenhouse gas measurements using the same type of analyzers, the authors tried to improve the water vapor corrections for CO2 and CO, and to determine the drift rate for the pressure and temperature sensors located inside the CRDS cavity. As the development presented here is some

sort of changes to or confirmation of the published methods and results, it is therefore in several places an overstatement for novelties. Furthermore, several methods and results are not (yet) convincing based on the presented results, see below.

I do agree with the second reviewer that the manuscript will benefit from English editing by a native speaker.

The authors tried to present a way of explaining the dependence of the $CO_2$ measurements on the flow rate, i.e. the outlet valve number, in Sect. 3.1. More information is needed to explain how Eq. 1 was derived. Was it derived from 2-point inlet pressure measurements? Appendix A gives a very nice theoretical analysis; however, I do not find it convincing to support the linear relation between actual $CO_2$ and raw $CO_2$. The equation apparently corrects the flow effect, which may actually reflect changes in something else, e.g. cavity temperature or pressure. Please show the raw measurement data to support this empirical equation.

Water vapor correction for CO: the authors rearranged (combined) the existing equations to fit a single equation to the experimental data. Statistically, the use of the 4000-data running mean should not change the results? Have the authors tried to fit the equation to the raw data?

With all the efforts to improve the water vapour equations, why did the author decide to include a cooler to dry the air?

The use of a large number of symbols makes it difficult to read. I would recommend simplifying them and showing only the necessary ones.

The Fortran 90 code does not make the work novel, and there is even no need to mention it in the main text.

---

## Author Comment (AC1) · 1 Sep 2018

**"Atmospheric CO2, CH4, and CO with CRDS technique at the Izaña Global GAW station: instrumental tests, developments and first measurement results" by Angel J. Gomez-Pelaez et al. ( https://www.atmos-meas-tech-discuss.net/amt-2017-375/ )**

**Author's Replies to the Comments of Referee #1**

This work describes the GHG measurement system at Izana Observatory (IZO). Several novel measurement methods or improvements to standard methods are described, including a new correction to CO2 in Picarro analyzers that is caused by inhomogeneous mixing in the measurement cell of the analyzer that is affected by flow rate. **It should be published after corrections outlined below.**

We acknowledge the constructive and extensive comments of the referee.

**R1.M1)** One major comment that should be addressed is the determination of the co2 correction for inlet pressure variation should be tested more. Details and recommendations are below.

We have performed additional tests to determine this correction. They confirm our previous findings. In the revised version of the manuscript we are going to detail these new tests and expand the information about the previous tests.

**R1.M2)** A second is that the authors use a few more complex fits and equations for corrections than reported previously in the literature for these same instruments, and there is little justification or evaluation of the improvement that these complex corrections make on the data, or reduction in error/uncertainty (e.g. quadratic fit to the CO2, and water vapour corrections).

In the new version of the manuscript we justify better and more these points mentioned. We agree with your comment that for CO2 we are using a more complex fitting in mole fraction than it is usually done in the literature (quadratic instead of linear), but this is not the case for the water vapour corrections. Concerning this H2O correction, for CO2 and CH4 we use equations of the same type than in the literature, whereas for CO we use almost the same equation than in the literature, indeed our equation is a bit simpler since one of the polynomials is cubic instead of quartic (see Eq. [5] of Chen et al., 2013, AMT amt-6-1031-2013; and section 2.2.5 of Karion et al. , 2013, AMT amt-6-511-2013). The novelties we present in the H2O correction for CO are in the way the processing is done. We are going to explain this more in the new version of the manuscript.

**Major & Minor comments:**

**R1.m1)** Acronyms: Some are introduced without definition in the abstract (CO2, WMO); Others esp. in the introduction are defined backwards from the usual convention, i.e. the acronym comes first with the expansion in parentheses. But not consistently. And some are introduced (WMO) prior to definition. The authors should carefully go through the text and fix these.

Thanks. These is fixed in the new version of the manuscript.

**R1.m2)** L29: awkward (requirements are higher as higher the lifetime...). Requirements are more stringent for gases with a longer lifetime perhaps?

Done as suggested.

**R1.m3)** P2, L4: American billion?

"American" removed. Now just "billion" (one thousand million). In the past, in British English one billion was one million million (this continues being the case in some languages).

**R1.m4)** P2, L4, ppm should be defined here as micromoles per mole of dry air, or dry mole fraction. (and similar for ppb).

We have added these suggestions.

**R1.m5)** P2 L8 awkward: "therefore, the required frequency... is much lower".

Done.

**R1.m6)** P2 L30 Tertiary shoudl not be capitalized I don't think.

Agreed.

**R1.m7)** P2 L32 until now

Done.

**R1.m8)** P3 L12 "being" should be ", rejecting the first hour as stabilization time."

Done.

**R1.m9)** Table 1 caption should say "for two averaging times". Some note should be made explaining the negative water vapour values here.

Done. Note that the precision for H2O measurements indicated by the manufacturer of the CRDS is: <200 ppm for 5-second averages, and < 50 ppm for 5-minute averages. Therefore, it is reasonable obtaining negatives values (-2.8 ppm) for H2O when measuring dry air, since, taking into account the precision of the instrument, these values are completely compatible with 0.0 ppm. A note clarifying this point is added.

**R1.m10)** Table 2 - wouldn't this be a more appropriate place to report the average H2O? It would be good to know if the water vapour had completely left the stream by the time these SD's were taken. Some explanation of the higher SD's over these 10 minutes compared to the Table 1 results should be made - perhaps the cylinder is still "stabilizing", but then did the authors see a trend in the measurement over the 10 minutes? Was any such trend or noise caused by water vapour still drying out?

We agree with your first comment. We are going to include in this table the average H2O. What any of the SD of tables 1 and 2 means is the SD of the sample of 1-minute, 10-minute or 60-minute means obtained along the full duration of the experiment (we clarify it better in the new version of the manuscript). We re-write your second comment as: the SD´s (associated to 10-minute means) in table 2 are larger than the SD´s in table 1 associated with 1-minute means. Apparently, this might seem contradictory since we expect a decrease in the random noise when increasing the averaging period. However, table 1 considers a 24-hour-long test, whereas table 2 considers a 72-hour-long test, and because of this, the SD of the set of 10-minute means is larger, since it includes the drift in the response of the instrument. Indeed, the purpose of this second test is monitoring the instrument drift along 3 days. We also include a note in the new version to point out this fact.

**R1.m11)** P4 L4: How was the ambient pressure test conducted? This paragraph should have more detail (pressure chamber?). I agree the effect of ambient pressure is small -you could note that this analyzer is not meant for use on aircraft where large pressure changes might occur. But I still wonder if this effect on the measurements is transient or dependent only on the absolute ambient pressure. Yver Kwok et al state that they find no ambient pressure dependence on CO2 and CH4.

In the new version of the manuscript we are including more details about this test. We will also add the note you mention. Thanks. We have not used a pressure chamber for the test, but simply taking into account the atmospheric pressure changes during the 72-hour test, in a similar fashion than ICOS-ATC (ICOS Atmospheric Thematic Centre for Atmosphere) did. Therefore, the test provides an upper-limit for the ambient pressure sensitivity, since there might be instrumental drift not attributable to atmospheric pressure changes. According to our knowledge, Yver Kwok et al. (2015) proceed in the same way than us (not using a pressure chamber). Our main purpose with this test was to discard we had one of the CRDS units whose CO measurements are affected by natural ambient pressure changes. We do think the sensitivities we have obtained for CO2 and CH4 are within what Yver Kwok calls not significant (we obtain 0.0038 ppm/hPa for CO2 and 0.047 ppb/hPa for CH4), as it happens for CO (0.04 ppb/hPa)

**R1.m12)** P4 L10: these pressures hould be given in millibar or SI units - presumably this is gauge pressure, psig. Absolute pressure would be better, perhaps stating the regulator gauge pressure in parentheses. Was the effect only noted for these two pressures, or was there more testing done at intermediate pressures? Was it repeatable, or are these numbers given only for the single test between two pressures? (this is a question I had later, on how the fit was made to the outlet valve - was the fit based only on two points, and was it evaluated for a repeated test?).

We agree, hPa are going to be used as units. Yes, this was gauge pressure. We are going to use absolute pressure following the referee´s advice. Since the effect is quite small, these two "extreme" pressures were used to maximize the signal to noise ratio. The test was repeated two different days with the same results. When accounting for this effect the fitting to the CO2 response function improved significantly in the first calibrations, when we were not so skilled adjusting precisely the CRDS inlet pressure corresponding to each gas cylinder. Therefore, intermediate pressures had been tested indirectly. The same considerations apply for the outlet valve.

We have performed additional tests to determine this correction, including intermediate pressure points and even a larger pressure range. They confirm our previous findings. In the revised version of the manuscript we are going to detail these new tests and expand the information about the previous tests.

**R1.m13)** p4 L23: "during that first calibration" - what calibration? perhaps "During the calibration described above,"

Done.

**R1.m14)** P4 L27, awkward phrasing "to the computation". Perhaps just "Pre-processing refers to the computation of raw data 3-second means... as well as the computation of some derived variables". (what variables?). Also "no" should read "not".

Done. Thanks.

**R1.m15)** P5 L3-10. It is not clear why certain variables are associated with certain species codes. I thought that the species code had to do with which laser was making the measurement (or the frequency range it was covering). One sentence explaining this perhaps would be useful to the reader. It might not be useful to mention the CO2_dry and CH4_dry variables, as they confuse things and they are not used in this work at all - they get updated after the h2o_reported scan. I find the explanation of each variable useful however. I think CO2_dry is erroneously mentioned twice: once with sepcies=2 and once when species=3.

Yes, you are right, the species code indicates in which spectral range the measurements are performed. There are 4 spectral ranges and 3 lasers. Two nearby spectral ranges are scanned with the same laser. We are going to add a sentence explaining it in the revised version of the manuscript. CO2_dry and CH4_dry are mentioned by completeness and for helping readers to "disentangling" the CRDS raw files (you are right, they are updated once the main spectral line of h2o is measured). The time sequence in which the species are scanned is going also to be indicated: 1, 2, 4 and 3. CO2_dry appears in two species: 2 and 3. This is not a mistake. The manufacturer might have decided two obtain an updated CO2_dry value using the H2O of the previous measurement cycle without waiting to the end of the current measurement cycle.

**R1.m16)** Why are both the MPV and SV fields used - should mention why these are monitored - presumably your gas handling uses both solenoid and MPV valves controlled by the Picarro software?

You are right, our gas handling system uses both solenoid and MPV valves controlled by the Picarro software. We are going to add a sentence mentioning it.

**R1.m17)** P5 L19, I believe the flow in the orifice itself is sonic, and is supersonic immediately downstream, depending on the ratio of upstream to cavity pressure. Also pressure units should be in mb or other SI, with torr in parentheses.

You are right, the flow has Mach number one just at the orifice, and larger than one just downstream de orifice. We are going to indicate it in the new version of the manuscript. As it is mentioned in Appendix A, within the inlet pressure range recommended by the manufacturer, the flow is always supersonic just after the orifice. We are going to use hPa units in the new version of the manuscript, as recommended by the referee.

**R1.m18)** P5 L20 was this linear relationship between OV and inlet pressure confirmed in your tests? This should be investigated and shown, not just assumed, as the inlet pressure/flow correction is using the OV variable as a direct proxy for flow rate essentially.

This linear relationship between OV and inlet pressure has been confirmed in numerous additional tests we have performed (using several pressures in each test). We are going to show this in the new version of the manuscript.

**R1.m19)** P5 Eq 1 - it would be good to see some more information on this fit. Was it really based only on the two different pressures investigated in the test? Could a fit to more pressures be performed? how good or bad is the fit - what are the residuals from the test itself? This is one of the main findings of this work, and has not been reported before, as the authors point out, so more work is needed to show the reader how the correction was determined, how it performs, and its consistency over time and after a reboot. The fit should

be tested by performing a test again and checking for consistency and for errors after the fit was applied, esp. its reliance on the OV variable.

Please, refer to our replies to comments R1.m12) and R1.m18).

I do believe that because the inlet orifice is not a physical orifice but rather a proportional valve fixed at a certain opening, after the analyzer has a power cycle (reboot), it may not come back to the same spot exactly (the valve is not so precise, so given the same voltage might open to a different diameter slightly), and following that, the OV may not be in the same spot, i.e. for the same inlet pressure as prior to the power cycle, the OV value may differ. I think that this would render equation 1 incorrect and would need to be re-calculated. This should be checked and determined. It may be necessary to install a pressure sensor upstream of the CRDS instrument and perform a linear fit on that rather than relying on the OV. This may not be true if there is a physical critical orifice upstream whose aperture never changes. The aircraft analyzers do have a physical critical orifice so that they can maintain a constant given flow rate, even after the instrument is rebooted.

According to the information provided to me by the manufacturer (Rella, private communication), "there is both a proportional valve and a physical orifice in the inlet system of the G2401. The proportional valve is opened slowly at startup to ensure that the flow smoothly changes, but after this startup procedure, the valve is set to full open, and the flow is set by the orifice".

Note also that if Eq. (1) is expanded, only the slope term in OV is kept fixed, since the independent term is, in practice, combined with the independent term of Eq. (6), and the latter equation is updated each time a mole fraction calibration is performed. Therefore, it automatically takes into account hypothetical drifts in the independent term of Eq. (1). The impact in the slope (in OV) by hypothetical drifts in the outlet valve controller is quite limited as it is going to be shown in the revised version of the manuscript.

The main advantage of the aircraft analysers is that they have the critical orifice at the outlet of the cavity instead of at the inlet. Since at critical orifices the (supersonic) flow rate is determined by the pressure and temperature at its inlet, for aircraft analysers the flow rate is constant no matter the huge changes in the pressure at the CRDS inlet that takes place during a flight.

**R1.m20)** p6 L22: perhaps rephrase to : "provided results that satisfied our accuracy requirements".

Done. Thanks.

**R1.m21)** P6 L24: This helps avoid...

Done.

**R1.m22)** P6 L25: that might propagate to the interior of the cylinders...

Done.

**R1.m23)** P7 L6: What reason is given for going to a quadratic fit for CO2 here? In other literature (including Yver Kwok et al) a linear fit is generally used. Perhaps the authors can justify the use of a more complex curve for this instrument. It is useful to note that the residuals of a calibration should be compared to the assigned 1-sigma uncertainty of the gas

standard relative to the scale. For NOAA tanks, this is something close to 0.02-0.03 ppm (as cited in Andrews et al., but could be checked for specific cylinders perhaps?). You wouldn't expect the residuals to be better than that.

No reason was provided in this part of the manuscript, but in the last paragraph of Sect. 2. In the new version of the manuscript we are going to justify also before Eq. (6) the reasons for using a quadratic fit.

**R1.m24)** P7 L6, L9: I would suggest replacing "real dry mole fraction" by "the dry mole fraction assigned by the CCL on the WMO scale".

Done. Also, after Eq. (5).

**R1.m25)** p7, L16 "high-accuracy"

Done. Since "accuracy" is the noun and "accurate" is the adjective, we preferred to use "high-accurate".

**R1.m26)** p7, L17 should read: the calibration fits are generally performed using a limited range...

Done. Thanks.

**R1.m27)** p7 Is the virtual tank concentration its assigned value (x-axis) or measured value (yaxis)? Presumably, following the convention of equations 4-6, it is the x-axis, assigned value that is constant at 400 ppm and the raw response for that value is what is calculated using the equations. Perhaps just state this in the text when you mention the tank values.

The virtual tank concentration is the assigned value (X-axis). Agreed, we are going to state what you mentioned in the text of the revised manuscript.

**R1.m28)** p8-9: explain that the drift you are referring to is drift in the P, T sensors that then causes the cavity to be controlled at a slightly drifting temperature. P9, L1: Was this supposed to read .152 degrees C / year rather than Torr?

Agreed, in the new version of the manuscript we are going to state explicitly what you mentioned. No, it is 0.152 degrees C/Torr since the first step is to obtain the slope in the p-T space using Eq. (8), and then it is possible to obtain the dp/dt value using Eq. (9). We are going to state it more clearly in the new version and provide also the dT/dt value.

**R1.m29)** Fig 2: is the slope of the red line here equal to the left hand side of equation 8?

When multiplying that slope by 400/1850, we obtain the left-hand side of Eq. (8) (see the last paragraph of page 7 in the Discussion paper). We are going to state it more explicitly in the new version of the manuscript.

**R1.m30)** Eqn 8 & 9: A little more information on this would be good. Does this analysis assume that Temperature drift is a linear function of pressure drift? I guess if they are both linearly drifting with time, this makes sense.

Equations (8) and (9) are general relationships between partial derivatives that do not rely on any assumption. They can be directly applied to characterize the response drifts in our paper since Figs. 1 and 2 have shown that the drifts are linear. We are going to state it explicitly in the new version of the manuscript. Since the left-hand-side of Eq. (8) is constant (linear relationship shown in Fig. (2)), the slope dT/dp is constant (due to Eq. (8)). Since the derivative

in time of the fractional change of CH4 is constant (linear relationship shown in Fig. 1), the slope dp/dt is constant (due to Eq. (9)), and therefore p drifts linearly in time. Finally, since dT/dp and dp/dt are constant, dT/dt is also constant, and therefore T drifts linearly in time.

What is the slope? (Yver Kwok report a mean of 2.4). Perhaps a plot of the fractional change rather tahn the plot in Figure 2 would be more applicable to explain equations 8 & 9 (similar to figure 14 in Y-K).

As it was stated at the end of page 7 of the Discussion paper, in our case the slope is 4.495, whereas Yver Kwok obtained a mean value of 2.4 for the set of CRDSs they tested. As mentioned in the response R1.m29, we are going to mention in the new version of the manuscript that both plots are tightly connected simply by multiplying: the X-values by 1/400 and the Y-values by 1/1850.

**R1.m31)** p9: The singular of species is species, so I think it should be "for each species" (here and throughout the paper).

Done. Thanks.

**R1.m32)** Fig 3: the Y-axis here is odd in terms of units. Not sure what to do about it.

You are correct, we think this is the best option because it is informative.

**R1.m33)** p11: L14: explain "since CO standards use to drift upward"? When?

What we mean here is that when a CO standard drifts, this drift is generally positive. We are going to indicate it explicitly in the revised version of the manuscript, and cite a few references, for example: https://www.esrl.noaa.gov/gmd/ccl/co_scale.html .

**R1.m34)** p11 L15: why at different rates? (or perhaps different rates from the target gas). If the 4 tertiary standards are drifting at different rates, one would expect residuals to be increasing in time on the fits as well? It seems that without further evidence, it might be more appropriate to just state that is is likely that the standards and/or target tanks are drifting.

Yes, this is exactly the case for CO as Fig. 4 shows: the calibration-fit RMS residual increases in time for CO. This is mentioned in the discussion manuscript a few lines before.

**R1.m35)** Fig4: the labels should be on the Yaxis rather than as titles. Not sure if this was stated earlier, but each of these points represents an average of how long and every how often? This could be mentioned in the caption to remind the reader.

Agreed, we are going to include the labels in the Y-axis. Each point represents the assigned value to a target gas in a mole-fraction calibration performed using 4 WMO laboratory standards (every 3 or 4 weeks); except the red dot that represents the calibration-fit RMS residual. We are going to include this information at the end of the caption: "and mole-fraction calibration performed using 4 WMO laboratory standards (every 3 or 4 weeks)".

**R1.m36)** Figure 5: I don't find this figure particularly useful. If kept, labels should be put in the photo of the various components.

Agreed. We are going to put labels in the photo of the various components.

**R1.m37)** Figure 6: y-axis is unlabeled, should have the variable even though there are not units on this raw measurement.

You are correct. We are going to label the y-axis using the variable and removing it from the interior of the plot.

**R1.m38)** p14 L24: 39 minutes seems to be very long, when looking at figure 6, the water vapour could change significantly in that time, especially at the higher H2O values at the beginning of the test. Is such a long average really required, when you are achieving 1-minute precision of 0.87 ppb?

The experiment lasted many hours. We have checked that using a 39-minute running mean for CO has no significant impact in the accuracy of the data for that experiment. Using a 39-minute running mean instead of a 1-minute running mean, the random noise is reduced by a factor 6 approximately.

Also, given the complex equations presented here for CO water correction, how different is the final result from the internal correction (i.e. the variable "CO" in the raw output)? It would be necessary for the reader to evaluate this and how it depends on the level of water vapour, to see if it's worth implementing.

Agreed. We are going to include a new figure in the revised manuscript to show it.

**R1.m39)** P15-16, Regarding the -40C trap, can you relate this to a maximum H2O value you see in the data after that point? What is the magnitude of the water correction at those values, and how different is it to CO2_dry and CH4_dry? Rella et al. indicate that at low water vapour values, the internal correction works well. Again, the water vapour tests are cumbersome, especially for field analyzers, in many cases - it would be useful to the readers to know what kind of improvement is gained from the tests as opposed to using the factory correction in cases where there already is partial drying like this.

During the first year in which our CRDS was in operation, no drying was performed. This made necessary the H2O correction. We are going to include a short paragraph in the revised manuscript providing the information you asked for when partial drying is performed.

**R1.m40)** P16, L7-8: Awkward phrasing "for not discarding". Perhaps "We retain a pre-processed 30-second mean if the following conditions are met: ". Then rephrase the conditions accordingly (i.e. "the mean values of the following variables are within...") etc.

Done.

**R1.m41)** L9: condition 2 awkward: was exit supposed to be "exist"?

You are correct. Thanks. Changed.

**R1.m42)** P16 L17: clarify when 10 minutes are discarded (after a calibration or any sample path switch?).

After any sample path switch. This is going to be explicitly indicated in the revised version of the manuscript.

**R1.m43)** P18 L14: remove the final "in space", as earlier this sentence stated in space and time.

Done. Thanks.

**R1.m44)** Fig 11, caption should include the fact that these are 12-hour averages (if that is the case).

Yes, this is the case. We are going to include this information in the caption.

**R1.m45)** Table 4, first row, month says 2017? I wonder if the information in this table would be better expressed in a figure somehow, with dashed lines indicating the WMO compatibility goals?

Indeed, the first row shows the mean values for the period 2015-2017. We need to change the table to make it clearer. If possible, we would prefer to keep the table.

**R1.m46)** P21 L4: shown

Done.

**R1.m47)** In section 6.3, the reason for the CRDS-RGA differences for CO is ascribed as due to the CRDS standard drift, but here in Section 7, it is indicated that the drift did not improve the comparison. Can the authors give any explanation for this? Even if the RGA tanks are also drifting, the correction described in Section 7 would put both data sets relative to the same tanks after all.

Section 6.3 mentioned that only as a hypothesis. We have no explanation for the results obtained in Section 7 concerning the CO comparison. Perhaps the explanation might be in the problems detected recently by the CCL in the CO WMO-X2014A scale (https://www.esrl.noaa.gov/gmd/ccl/co_scale_update.html ).

**R1.m48)** Finally, overall, given the large effort involved in achieving very high accuracy for this data set, do the authors envision prescribing an uncertainty to their measurements? [perhaps this is outside the scope of this paper, but something to think about maybe for the future].

That is outside the scope of this paper. Thanks for suggesting it as something to think about for the future.

---

## Author Comment (AC2) · 1 Sep 2018

**"Atmospheric CO2, CH4, and CO with CRDS technique at the Izaña Global GAW station: instrumental tests, developments and first measurement results" by Angel J. Gomez-Pelaez et al. ( https://www.atmos-meas-tech-discuss.net/amt-2017-375/ )**

**Author's Replies to the Comments of Referee #2**

In this paper, the authors present their instrumental set-up to measure greenhouse gases continuously at Izana global GAW station. The instrument performances are tested and interferences are studied. First ambient air measurements are compared against the historical NDIR and GC systems. **This paper should be published after corrections as detailed below**.

We acknowledge the comments of the referee.

General comments:

**R2.M1)** The paper would benefit from an English proofing.

This is going to be done before submitting the revised version of the manuscript.

Specific comments:

**R2.m1)** p4: It would be great to have a plot of the inlet pressure tests. Did you only varied between two pressures? Also, it would be good to make this test several time to see if drifts occurs over time and if the results are reproducible as you are using them to correct data.

Since the effect is quite small, two "extreme" pressures were used to maximize the signal to noise ratio. The test was repeated two different days with the same results. When accounting for this effect the fitting to the CO2 response function improved significantly in the first calibrations, when we were not so skilled adjusting precisely the CRDS inlet pressure corresponding to each gas cylinder. Therefore, intermediate pressures had been tested indirectly.

We have performed additional tests to determine this correction, including intermediate pressure points and even a larger pressure range. They confirm our previous findings. In the revised version of the manuscript we are going to detail these new tests and expand the information about the previous tests. Also, we are going to include a plot as requested by the referee.

**R2.m2)** p4: You mention using smaller inlet pressures differences, does that lead to smaller RMS residuals? Is then the inlet pressure correction useful?

Being able to get smaller inlet pressures differences decreases the impact of the pressure correction, of course. Keeping small the inlet pressure differences between ambient air and standards is quite difficult. We do think that taking into account the pressure correction keeps us in the safe-side to obtain high-accuracy CO2 measurements.

**R2.m3)** p5 l25: From experience, the outletvalve value change with time, depending on the Tdas temperature, after restarting the instrument or when the filters get clogged. It seems safer to operate as you said yourself by reducing the difference in inlet pressure between cylinders and ambient air to avoid the need for an empirical correction.

Note that if Eq. (1) is expanded, only the slope term in OV is kept fixed, since the independent term is, in practice, combined with the independent term of Eq. (6), and the latter equation is updated each time a mole fraction calibration is performed. Therefore, it automatically takes into account hypothetical drifts in the independent term of Eq. (1). The impact in the slope (in OV) by hypothetical drifts in the outlet valve controller is quite limited as it is going to be shown in the revised version of the manuscript.

**R2.m4)** Section 5: did you perform more than one test to assess the variability? It would be interesting to plot the biases between the assigned dry value and the wet values depending on $H_2O$ for Picarro and your own correction for the three species. On the same subject: p15 l22: what is the level of residual water? Why not invest in a -60 C cryocooler and get rid of any correction as you are using already a crycooler system? Especially if you refer yourself to the paper of Reum et al. in discussion in AMT (Reum, F., Gerbig, C., Lavric, J. V., Rella, C. W., and Göckede, M.: An improved water correction function for Picarro greenhouse gas analyzers, Atmos. Meas. Tech. Discuss., https://doi.org/10.5194/amt-2017-174, in review, 2017.) that shows that the $H_2O$ correction is biased when almost dry due to the sensitivity of the pressure sensor with $H_2O$.

We performed a few, but kept only this one due to its superior design and performance. We are going to include the plots you mentioned in the revised manuscript. During the first year in which our CRDS was in operation, no drying was performed. This made necessary the $H_2O$ correction. We are going to include a short paragraph in the revised manuscript providing the information you asked for when partial drying is performed. We had a -40 C cryocooler available. Funds are always scarce.

Concerning the Discussion paper you mentioned (Reum et al., 2017), the AMT website indicates: "This discussion paper is a preprint. It has been under review for the journal Atmospheric Measurement Techniques (AMT). The revised manuscript was not accepted". Therefore, we can not use the results of that manuscript.

**R2.m5)** Section7: Can the ambient air difference be due to the non-linearity of the RGA-3 or to the fact that the $H_2O$ correction is not good enough? The difference does not seem to increase strongly over time but more to vary around a bias.

We do think the non-linearity of the RGA-3 is well characterized and is taken into account. We discard this might be due to the $H_2O$ correction: the amount of $H_2O$ has been very small since November 2016 due to the -40 C cryocooler. We have no explanation for the bias between the CO measurements you mentioned. Perhaps the explanation might be in the problems detected recently by the CCL in the CO WMO-X2014A scale (https://www.esrl.noaa.gov/gmd/ccl/co_scale_update.html ).

Technical comments:

**R2.m6)** p1 l29: not clear, please rephrase

Done.

**R2.m7)** p2 l7 replace "being.. much lower" by reducing

Re-phrased following the indications of Referee 1.

**R2.m8)** p2 l3 replace "ones" by "techniques"

Done.

**R2.m9)** p2 l26: add "of" after "physical discussion"

Done.

**R2.m10)** p2 l27: "as follow"

According to the WordReference dictionary, it is correct written as "as follows".

( http://www.wordreference.com/es/translation.asp?tranword=as%20follows )

**R2.m11)** p3 l6 "serial number" should go before the actual serial number

Done.

**R2.m12)** p3 l7: remove "to the CRDS"

We do think it is necessary to keep it in order to have a sentence with full sense.

**R2.m13)** p3 l9: cite as well here Yver Kwok et al. As the tests are described in this paper while the specifications gives the tresholds.

Indeed, the mentioned ICOS-ATC report also describes the recommended tests in the year in which it was published (2016), whereas Yver Kwok included all the CRDS tested along the years since 2008. Along these years the methods were not kept constant but they evolved. Therefore, we are going to re-write the sentences to state clearly that the report also describes the test method.

**R2.m14)** p3 l11: In Yver Kwok et al. the terminology for the precision test is continuous measurement repeatability (CMR), rephrase for example as "The first test, defined in Yver Kwok et al; as the CMR test consists..."

In the revised manuscript, we are going to include the sentence: "(this test is called continuous measurement repeatability -CMR- by Yver Kwok et al., 2015)". So, by the way, we are going to cite Yver Kwok et al., 2015, here.

**R2.m15)** p3 l12 put "being" after "the first hour". This type of exchange appears throughout the text, please check.

We have changed it following the advice of Referee 1.

**R2.m16)** p3 l15: Replace precision by CMR

We are going to include both terms.

**R2.m17)** p3 l18: this test is what was called reproductibility before and in ICOS-ATC LTR or long term repeatability. Add ATC after ICOS as ecosystem or ocean could have defined other terms.

In the revised manuscript, we are going to include the sentence: "(this test is called long-term repeatability -LTR- by Yver Kwok et al., 2015)", and to add ATC after ICOS.

**R2.m18)** p3 l24: thresholds

Done.

**R2.m19)** p4 l2 Replace "Repeatability test" by LTR

We are going to include both terms.

**R2.m20)** p4 l5 use proper units, mbar not mb

You are correct. We are going to use hPa (= mbar).

**R2.m21)** p4 27-28 delete "to" before "the computation"

We have re-written the sentence as advised by Referee 1.

**R2.m22)** p6 l2 delete "tot" before "the raw CH4"

According to an Oxford dictionary, the usual structure is: "call somebody/something + noun". However, we are using "call noun to somebody/something" since in the other way the sentence might be confusing because the used "somebody/something" is quite large. We are going to check it during the English proofing.

**R2.m23)** p6 l12-13 you multiply p84sw by 1000, then p84sw' should be in ppb contrary to what is mentioned.

Please, look Eq. (5) and the last plot of Figure 3. The coefficient $b$ is approximately equal to 2.32. This explains why p84sw' has no ppb units but it is a raw value.

**R2.m24)** p6 l28: numerical

Done.

**R2.m25)** p7 l28 It is a new paragraph and the transition is not smooth, please add a transition.

We are going to write it as a new paragraph and make the transition smoother.

**R2.m26)** p11 l15 Refer to section 7 for more investigations.

Agreed.

**R2.m27)** p14 l 7-8: move "being" after "CH4", add "s" to coefficient

Done.

**R2.m28)** p15 l7 what is pph?

This was defined in p13 l3-4: "parts per hundred in mole fraction". Now it is complemented with: "i.e., millimoles per mole of air".

**R2.m29)** p15 l16: why is there two inlets, what is the purposes of switching between the two every day?

This has two purposes: 1) to provide a lot of time to change the flask used to trap $H_2O$ in the air line not used at this moment; 2) to check the consistence between both lines after every switching: a bias between them might indicate a leak in one of the lines (especially in the general inlet, which has a few large unions and several instruments connected to it; or at the flasks connections). We are going to add this explanation in the revised manuscript.

**R2.m30)** p16 l8: Rephrase "For not discarding" in "To keep"

We have re-written it following the advice of Referee 1.

**R2.m31)** p18 l18: Can you comment on the reason of the larger differences for CO2 and CH4 for these particular months? As it is both for CO2 and CH4, it seems to indicate that the CRDS would be the cause?

The larger differences took place in October and November 2016. We do think they were produced by a small leak in the general inlet used by the CRDS (the NDIR and the GC-FID used another general inlet). We are going to add this explanation in the revised manuscript.

**R2.m32)** p20 l10: Replace "for comparing" by "to compare"

Done. Thanks.

---

## Author Comment (AC3) · 1 Sep 2018

**"Atmospheric CO2, CH4, and CO with CRDS technique at the Izaña Global GAW station: instrumental tests, developments and first measurement results" by Angel J. Gomez-Pelaez et al. ( https://www.atmos-meas-tech-discuss.net/amt-2017-375/ )**

**Author's Replies to the Comments of Referee #3**

Gomez-Pelaez et al. present both laboratory and field test results of a commercially available CO2/CH4/CO cavity ring-down spectrometer (CRDS). The authors discussed the results within the context of several relevant international programs, e.g. Global Atmosphere Watch (GAW) and the European Integrated Carbon Observation System (ICOS). Being aware of recent development of greenhouse gas measurements using the same type of analyzers, the authors tried to improve the water vapor corrections for CO2 and CO, and to determine the drift rate for the pressure and temperature sensors located inside the CRDS cavity.

**R3.M1)** As the development presented here is some sort of changes to or confirmation of the published methods and results, it is therefore in several places an overstatement for novelties. Furthermore, several methods and results are not (yet) convincing based on the presented results, see below.

We disagree with this comment.

**R3.M2)** I do agree with the second reviewer that the manuscript will benefit from English editing by a native speaker.

This is going to be done before submitting the revised version of the manuscript.

**R3.M3)** The authors tried to present a way of explaining the dependence of the CO2 measurements on the flow rate, i.e. the outlet valve number, in Sect. 3.1. More information is needed to explain how Eq. 1 was derived. Was it derived from 2-point inlet pressure measurements? Appendix A gives a very nice theoretical analysis; however, I do not find it convincing to support the linear relation between actual CO2 and raw CO2. The equation apparently corrects the flow effect, which may actually reflect changes in something else, e.g. cavity temperature or pressure. Please show the raw measurement data to support this empirical equation.

Since the effect is quite small, two "extreme" pressures were used to maximize the signal to noise ratio. The test was repeated two different days with the same results. When accounting for this effect the fitting to the CO2 response function improved significantly in the first calibrations, when we were not so skilled adjusting precisely the CRDS inlet pressure corresponding to each gas cylinder. Therefore, intermediate pressures had been tested indirectly.

We have performed additional tests to determine this correction, including intermediate pressure points and even a larger pressure range. They confirm our previous findings. In the revised version of the manuscript we are going to detail these new tests and expand the information about the previous tests. We also are going to show raw measurements data to support the empirical equation, as requested by the referee.

Concerning the referee´s sentence: "The equation apparently corrects the flow effect, which may actually reflect changes in something else, e.g. cavity temperature or pressure", we disagree. The CRDS controls "perfectly" the temperature and pressure of the cavity, with very short transient periods, and these variables are continuously monitored by the CRDS and recorded in the acquired raw data files. Temperature or pressure drifts in the cavity sensors can only have an effect on long-term periods (several months or years). The flow effect due to changes in the inlet pressure affects on short timescales, due to the switching between different "air sources" (ambient air, laboratory standards, and target gas) at slightly different pressures.

**R3.M4)** Water vapor correction for CO: the authors rearranged (combined) the existing equations to fit a single equation to the experimental data. Statistically, the use of the 4000-data running mean should not change the results? Have the authors tried to fit the equation to the raw data?

The $H_2O$ experiment lasted many hours. We have checked that using a 39-minute running mean for CO has no significant impact in the accuracy of the data for that experiment. Using a 39-minute running mean instead of a 1-minute running mean, the random noise is reduced by a factor 6 approximately. Least-squares fitting is very sensitive to outlayers (they have a large weight in the computation of the coefficients to be determined). Therefore, the smoothing of the CO data (in a safe way) through a running mean makes the fit more robust and the determined coefficients have less uncertainty.

**R3.m1)** With all the efforts to improve the water vapour equations, why did the author decide to include a cooler to dry the air?

Till we got all the materials necessary to install the final plumbing configuration detailed in Sect. 6, we needed to use during one year a simpler configuration without drying. During the first year in which our CRDS was in operation, no drying was performed. This made necessary the $H_2O$ correction.

**R3.m2)** The use of a large number of symbols makes it difficult to read. I would recommend simplifying them and showing only the necessary ones.

Almost all the symbols shown are necessary. The few ones remaining are shown by completeness and for helping readers to "disentangling" the CRDS raw files

**R3.m3)** The Fortran 90 code does not make the work novel, and there is even no need to mention it in the main text.

We are going to remove the words "Fortran 90" in the main text, keeping them only at the end of the manuscript in the paragraph called "Code availability".

---

## Referee Report (RR1)

Review of version 2:
"Atmospheric CO2, CH4, and CO with CRDS technique at the Izaña Global GAW station: instrumental tests, developments and first measurement results", by Angel J. Gomez-Pelaez et al.

Over all, the authors have made a commendable effort at addressing previous reviewer comments, specifically by conducting a series of additional tests to illustrate the dependence of CO2 measurements on the analyzer inlet pressure. Much of the work and equations have been clarified. My main additional comment is on grammar - the document has many awkwardly phrased sentences. I have pointed out some of them below, but it would benefit from a thorough grammatical edit/review for language.

Specific comments:

Section 2.3 (P4 L20 and on):
A brief one or two sentence description of this test should be given here so the reader does not have to refer to Yver Kwok. Otherwise it is not clear how this test is different from the one described in 2.4. I believe what is referred to here is keeping a constant inlet pressure and flow rate but allowing the room pressure (ambient pressure) to vary.

P4 L 25 - can the authors provide a reference as to what they are referring to here - are there certain CRDS units that have a dependence of CO on natural ambient changes? Otherwise, one could be more general, as in "our main purpose was to confirm that the CRDS unit's CO2, CH4, or CO measurements are not affected by natural ambient pressure changes."

P5 section heading is a partial sentence but has a period (grammar). I think just replacing the period with a colon (:) would work. Also first sentence grammar: "its" should be "their", as it is describing "relationships".

Thank you for the complete description here of choked flow. I did not know that the non-aircraft units also used a critical orifice.

P5 L28: should be "downstream of the CRDS, but upstream of the vacuum pump".

This section is much improved, with the additional tests making a much more robust point about the influence of inlet pressure (or flow rate) on the measurement for CO2. This finding likely has implications for other users of these analyzers if they are not controlling pressure upstream to ensure the same flow rate between sample and standards.

P12 L6-10 sentence is hard to follow

P13 eq 8,9: Wouldn't it be easier to characterize the calibration with the CO2 from the standards on the Y-axis and have CO2ovc be the dependent (x) variable? Then solving the quadratic equation is not needed. Same could be said for equations 6 & 7, I would

think this is the more common format.  Regardless, no need to address this, it just seems it would be easier on the computation.

P24, L9 consistence should be consistency

P24 L12: should just say water vapor correction or H2O correction?

P25 L1-2 grammar

P32 L22: Can the authors be more quantitative here about what "good" means?  The CRDS passed the acceptance tests for ICOS?

P33 L25: should read "we have determined and discussed the physical origin of…". This occurs elsewhere in the text as well, where "physical" is used awkwardly.

---

## Author Response (AR2)

**"Atmospheric CO2, CH4, and CO with CRDS technique at the Izaña Global GAW station: instrumental tests, developments and first measurement results" by Angel J. Gomez-Pelaez et al. ( https://www.atmos-meas-tech-discuss.net/amt-2017-375/ )**

**Associate Editor Decision: Publish subject to minor revisions (review by editor) (17 Jan 2019)**

**Author´s Replies to the Comments of Associate Editor**

**E1)** Please prepare a revised version of your paper by taking in account the suggestions of Referee #3 (report #1) and the recommended technical correction of Referee #1 (report #2).

Done.

**E2)** In addition, I kindly ask you to correct the manuscript for English, as you promised for this next version of the manuscript.

The English language has been revised by a native speaker.

**Author's Replies to the Comments of Referee #1**

**R1.M1)** Over all, the authors have made a commendable effort at addressing previous reviewer comments, specifically by conducting a series of additional tests to illustrate the dependence of CO2 measurements on the analyzer inlet pressure. Much of the work and equations have been clarified.

Thanks for appreciating the work carried out.

**R1.M2)** My main additional comment is on grammar – the document has many awkwardly phrased sentences. I have pointed out some of them below, but it would benefit from a thorough grammatical edit/review for language.

The English language has been revised by a native speaker.

**R1.m1)** Section 2.3 (P4 L20 and on): A brief one or two sentence description of this test should be given here so the reader does not have to refer to Yver Kwok. Otherwise it is not clear how this test is different from the one described in 2.4. I believe what is referred to here is keeping a constant inlet pressure and flow rate but allowing the room pressure (ambient pressure) to vary.

We have added the following sentence: "inlet gauge pressure was kept constant but the lab pressure (ambient pressure) was allowed to naturally vary". Note that the ambient pressure changes are quite small compared with the inlet pressure changes considered in Sect. 2.4.

**R1.m2)** P4 L 25 - can the authors provide a reference as to what they are referring to here – are there certain CRDS units that have a dependence of CO on natural ambient changes? Otherwise, one could be more general, as in "our main purpose was to confirm that the CRDS unit's CO2, CH4, or CO measurements are not affected by natural ambient pressure changes."

Yes. We have added at the end of the sentence: "as found Yver Kwok et al. (2015) for several units" (in detail, at the end of its Sect. 4.3).

**R1.m3)** P5 section heading is a partial sentence but has a period (grammar). I think just replacing the period with a colon (:) would work. Also first sentence grammar: "its" should be

"their", as it is describing "relationships". Thank you for the complete description here of choked flow. I did not know that the non-aircraft units also used a critical orifice.

Done. Thanks.

**R1.m4)** P5 L28: should be "downstream of the CRDS, but upstream of the vacuum pump". This section is much improved, with the additional tests making a much more robust point about the influence of inlet pressure (or flow rate) on the measurement for CO2. This finding likely has implications for other users of these analyzers if they are not controlling pressure upstream to ensure the same flow rate between sample and standards.

Done. Thanks.

**R1.m5)** P12 L6-10 sentence is hard to follow

We have rewritten the sentence as follows:

"In Appendix A, we provide a plausible physical explanation for this effect, **after determining the H2O dependence of the CRDS flow and arguing that** the pressure field inside the CRDS cavity is slightly inhomogeneous. Note that our arguments that point out to the fact that the inhomogeneity of the pressure field inside the cavity due to the flow produces the CO2raw dependence on OV, seem to be also supported by the following fact: the relative effect of the flow on CO2raw and CH4raw is roughly the same, *as we could roughly expect from the decrease in trace gas concentration (mol per volume) that takes place inside the CRDS cavity near the inlet and outlet, due to the decreased pressure (see Appendix A).* Note that dividing the slopes of Table 5 by the typical ambient air mole fraction at IZO (405 ppm for CO2 and 1880 ppb for CH4) we obtain: 1.06·10-8 for CO2 and 1.31·10-8 for CH4."

**R1.m6)** P13 eq 8,9: Wouldn't it be easier to characterize the calibration with the CO2 from the standards on the Y-axis and have CO2ovc be the dependent (x) variable? Then solving the quadratic equation is not needed. Same could be said for equations 6 & 7, I would think this is the more common format. Regardless, no need to address this, it just seems it would be easier on the computation.

For our GHG measurement systems at Izaña Observatory, we use to consider the relationships in this way since sometimes there are more than one independent variable, whereas the response of the instrument is the true dependent variable. For example, consider an instrument whose response does not only depend on mole fraction but also on temperature and pressure.

**R1.m7)** P24, L9 consistence should be consistency

Done. Thanks.

**R1.m8)** P24 L12: should just say water vapor correction or H2O correction?

Done. Thanks.

**R1.m9)** P25 L1-2 grammar

Rewritten.

**R1.m10)** P32 L22: Can the authors be more quantitative here about what "good" means? The CRDS passed the acceptance tests for ICOS?

We have replaced the mentioned sentence with "The CRDS passed the acceptance tests", without providing quantitative details that would have required too much space for this final section. Also, we prefer to avoid using here the sentence "for ICOS", since the tests were not performed by ICOS and as we indicate in Sect. 2, we have roughly followed the recommendations provided by ICOS because Izana station is not part of ICOS.

**R1.m11)** P33 L25: should read "we have determined and discussed the physical origin of…". This occurs elsewhere in the text as well, where "physical" is used awkwardly

Done. We have revised also the use of "physical" in the rest of the manuscript.

**Author's Replies to the Comments of Referee #3**

**R3.m1)** This work is intended to be published as a scientific paper. In my opinion, it should not be so eager to claim its "novelties", especially when it is not scientifically sound. In the abstract, line 13, "through novel numerical codes" is not relevant. Instead of claiming the "novelties", I suggest that the authors describe briefly (summarize) what have been exactly done, and what the results are in plain scientific languages. For example, instead of "novelties in the determination of the $H_2O$ correction for CO", explain what method has been applied; mention what is the origin of the CRDS-flow inlet pressure and $H_2O$ dependences. This is not to downplay the contribution of this work, but detailed justification of the use of virtual tanks should not be considered novel at all.

Done.

**R3.m2)** Reum et al., are in review (https://www.atmos-meas-tech-discuss.net/amt-2018-242/) as a separate submission. It would be useful to add discussions on the reported sensitivity of the pressure sensor with $H_2O$, since both this work and Reum et al. point to potential biases in cavity pressures. The referee´s original sentence: "The equation apparently corrects the flow effect, which may actually reflect changes in something else, e.g. cavity temperature or pressure" meant to indicate this sensitivity of the pressure sensor with $H_2O$ reported in Reum et al., but it may not be made clear in the original review.

We have added two paragraphs in Sect. 5 discussing briefly the results of Reum et al.

[revised manuscript text omitted]